Review Article

# Sepsis-induced changes in pyruvate metabolism: insights and potential therapeutic approaches

Louise Nuyttens [1,2], Jolien Vandewalle [1,2] & Claude Libert [1,2]✉

## Abstract

**Sepsis is a heterogeneous syndrome resulting from a dysregulated host response to infection. It is considered as a global major health priority. Sepsis is characterized by significant metabolic perturbations, leading to increased circulating metabolites such as lactate. In mammals, pyruvate is the primary substrate for lactate production. It plays a critical role in metabolism by linking glycolysis, where it is produced, with the mitochondrial oxidative phosphorylation pathway, where it is oxidized. Here, we provide an overview of all cytosolic and mitochondrial enzymes involved in pyruvate metabolism and how their activities are disrupted in sepsis. Based on the available data, we also discuss potential therapeutic strategies targeting these pyruvate-related enzymes leading to enhanced survival.**

**Keywords** Sepsis; Pyruvate; Lactate; Metabolism; Mitochondria
**Subject Categories** Metabolism; Microbiology, Virology & Host Pathogen Interaction

## Sepsis: general introduction

Sepsis is defined as a life-threatening organ dysfunction caused by a maladaptive host response to infection (Singer et al, 2016). Globally, it has a yearly incidence of 49 million people resulting in 11 million deaths (22% of all cases), constituting almost 20% of the total worldwide mortality (Rudd et al, 2020). Above this, sepsis is also the main cause of morbidity and mortality in intensive care units (ICUs). Hence, the World Health Organization (WHO) labeled sepsis as one of the most urgent unmet medical needs of today (Reinhart et al, 2017). Despite five decades of research, effective bench-to-bedside treatments targeting mechanistic pathways and reducing mortality rates are still lacking. Current treatments are rather supportive than curative and mainly consist of antibiotics, blood pressure control, and organ support (Van Wyngene et al, 2018). Unfortunately, sepsis prevalence is expected to rise over the years due to the global aging of the population and the occurrence of antibiotic resistance.

Classically, the pathophysiology of sepsis is thought to consist of a primary, pro-inflammatory response to eliminate the invading pathogens, followed by an immunosuppressive response, considered important to induce tissue repair (Hotchkiss et al, 2016). Consequently, many clinical trials have focused on testing anti-inflammatory drugs, immunomodulating agents and even pro-inflammatory molecules, but none of them had a beneficial impact on sepsis outcome (Van Der Poll et al, 2017). Therefore, it has become increasingly clear that sepsis is not solely an inflammatory syndrome, but rather a complex and heterogeneous condition, involving other major pathological changes beyond inflammation, such as changes in microbiome composition, coagulation, complement activation, thermoregulation, circadian rhythm, and metabolic alterations (Cohen et al, 2015). The latter is especially important, given that the latest definition of septic shock (Sepsis-3) associates metabolic abnormalities with a higher mortality rate (Singer et al, 2016). Indeed, many studies already highlighted the presence of significant metabolic alterations accompanied with a failing starvation response (Detailed explanation in Box 1; Van Wyngene et al, 2018; Vandewalle and Libert, 2022). This is associated with elevated circulating metabolites such as lactate.

Lactate is a well-known biomarker of illness severity and a strong predictor of mortality in sepsis (Suetrong and Walley, 2016). Blood lactate levels of 1–2 mM are considered normal. Patients with septic shock show increasing lactate levels from 4 mM up to 20 mM and have a very poor prognosis with a mortality rate of >40% (Ryoo et al, 2018; Lee et al, 2021). Hence, lactate is thought to play a versatile role in sepsis (Detailed explanation in Box 2). The precise mechanism of sepsis-induced hyperlactatemia remains enigmatic. While some studies propose that reduced hepatic clearance may play a role, the primary factor driving high lactate presence may also be increased lactate production (Suetrong and Walley, 2016). Indeed, besides the failing starvation response, sepsis subjects are characterized by a metabolic switch towards increased glycolysis to provide energy to compensate for impaired mitochondrial functioning (Van Wyngene et al, 2018). This sepsis-related mitochondrial dysfunction is assumed to be caused by cytopathic hypoxia as many studies have shown that septic tissues have an impaired ability to use oxygen, despite normal oxygen levels in the blood. Hence, oxidative phosphorylation (OXPHOS)-dependent ATP production becomes restricted and aerobic glycolysis is increased as an alternative, but less efficient source of ATP. With this, pyruvate is reduced to lactate by lactate dehydrogenase (LDH)

[1]Center for Inflammation Research, Vlaams Instituut voor Biotechnologie (VIB), Ghent, Belgium. [2]Department of Biomedical Molecular Biology, Ghent University, Ghent, Belgium. ✉E-mail: claude.libert@irc.vib-ugent.be

**Glossary**

| | | | |
|---|---|---|---|
| **Aerobic glycolysis (Warburg effect)** | The generation of ATP via increased rates of glycolysis rather than OXPHOS despite the presence of adequate oxygen levels and fully functioning mitochondria. This was first described by Otto Warburg as a hallmark of cancer cells. | | glucose-1-phosphate and subsequently into glucose-6-phosphate to enter the glycolysis pathway. |
| | | **Glycolysis** | The metabolic process involving the breakdown of glucose into pyruvate to generate ATP, NADH and $H^+$. |
| **Anaplerosis** | Metabolic pathways that replenish intermediates needed for critical biological reactions (e.g., TCA cycle intermediates). | **Lipolysis** | The enzymatic breakdown of lipids into FFAs and glycerol. |
| **Beta-oxidation** | The catabolic process of FFA breakdown into Acetyl-CoA. | **Lipotoxicity** | Lipid accumulation in non-adipose tissues that results in cellular dysfunction and death. |
| **Cecal ligation and puncture (CLP)** | The golden standard model of sepsis (usually in mice or rats) as it provides a better representation of the complexity of human sepsis. It encompasses a combination of three insults: tissue trauma due to laparotomy, necrosis caused by ligation of the cecum, and infection due to the leakage of peritoneal microbial flora into the peritoneum. Altogether, this results in polymicrobial peritonitis (Dejager et al, 2011). | **Oxidative phosphorylation (OXPHOS)** | The metabolic process involving series of oxidation-reduction reactions finally resulting in the reduction of oxygen and the generation of ATP. |
| | | **Proteolysis** | The enzymatic breakdown of proteins into amino acids. |
| **Cytopathic hypoxia** | Diminished rate of oxygen consumption by the mitochondria despite normal $PO_2$ values in the blood. | **Starvation response** | A set of metabolic adaptions in response to reduced energy intake. This involves the mobilization of energy reserves such as glycogen, fat and proteins through the catabolism of body tissue (adipose tissue and muscle) |
| **Endotoxemia** | The presence of endotoxins (e.g., LPS) in the blood. LPS injection, i.e., endotoxemia model, is used to mimic the systemic inflammatory response linked with early sepsis. | **Steatosis** | Fat accumulation in the liver. |
| | | **Sepsis and septic shock (Sepsis-3)** | In 2016, the Society of Critical Care Medicine and the European Society of Intensive Care Medicine redefined **sepsis** as a life-threatening organ dysfunction resulting from a dysregulated host response to infection. With this, **septic shock** was revised as a subset of septic individuals with increased circulatory, cellular, and metabolic abnormalities, i.e., refractory hypotension (mean arterial pressure ≤65 mm Hg) requiring vasopressors and serum lactate levels greater than 2 mmol/L (Singer et al, 2016). |
| **Free fatty acids (FFAs)** | Non-esterified fatty acids that are produced by the hydrolysis of triglycerides in adipose tissues. FFAs are a major fuel source for many tissues in the body and can be converted into ketone bodies by the liver. | | |
| **Gluconeogenesis (GNEO)** | The metabolic process involving the production of glucose from smaller precursors, such as glycerol, amino acids and lactate. | | |
| **Glycogenolysis** | The enzymatic process involving the breakdown of glycogen, an energy reserve found in the liver and skeletal muscle, into | **Tricarboxylic acid (TCA) cycle** | Also known as the Krebs cycle, occurs in the mitochondria and is a cyclic series of biochemical reactions resulting in the |

thereby regenerating $NAD^+$ from NADH for maintaining glycolysis activity (Fig. 1) (Van Wyngene et al, 2018). Given that pyruvate is the primary substrate for LDH activity to produce lactate (in mammals) and that it is mainly produced by glycolysis and consumed by mitochondria, it is logical that some studies have shown a disturbed pyruvate metabolism in sepsis. This involves changes in activity of enzymes and carriers regulating pyruvate metabolism, eventually playing a pivotal role in sepsis lethality, as will be discussed in detail within this review. Hence, this review will provide an in-depth exploration of the alterations in pyruvate metabolism that have been observed in sepsis (models). We will also provide an overview of the potential therapeutic options addressing these perturbations and how they improve sepsis survival.

# Pyruvate metabolism: general introduction

Pyruvate is the end-product of glycolysis. It serves as a pivotal, metabolic node in cellular metabolism as it links glycolysis with the tricarboxylic acid (TCA) cycle and OXPHOS pathway to generate ATP. In normal eukaryotic cells, glycolysis is the main route of pyruvate production, where the final step involves the conversion of phosphoenolpyruvate (PEP) into pyruvate by pyruvate kinase (PK). In the cytosol, the minority of the formed pyruvate can be reversely converted into either lactate by LDH or alanine by alanine aminotransferase (ALT). However, the majority of the cytosolic pyruvate is predominantly transported across the inner mitochondrial membrane by the mitochondrial pyruvate carrier (MPC) into the mitochondrial matrix. Depending on the energy status of the cell, the transported pyruvate can either be oxidized by pyruvate dehydrogenase complex (PDC) to form carbon dioxide ($CO_2$) and acetyl-CoA, which can be further oxidized by the TCA cycle for mitochondrial ATP generation, or used for fatty acid synthesis. Alternatively, the transported pyruvate can be carboxylated by pyruvate carboxylase (PC) to generate oxaloacetate which can either drive the TCA cycle, fatty acid synthesis or it can be used in pyruvate-driven gluconeogenesis (GNEO). Another important route for pyruvate metabolism involves malic enzyme (ME), which catalyzes the oxidative decarboxylation of malate to pyruvate and $CO_2$ (Gray et al, 2014; Jeoung et al, 2014; Prochownik and Wang, 2021) (Fig. 2).

**Box 1  Metabolic alterations and failing starvation response in sepsis**

Hallmarks of sepsis include immune activation, phagocytosis, acute phase reactant production, fever, tachycardia, and tachypnea, which all require increased supraphysiological energy supplies (Van Wyngene et al, 2018; Wasyluk and Zwolak, 2021). Despite this, septic patients are often unwilling or unable to consume food and exhibit mitochondrial dysfunction, resulting in a reduced capacity of cells to produce ATP (Peterson et al, 2010; Wang et al, 2016; Singer and Brealey, 1999). Muscle biopsies from septic patients are characterized by a diminished ATP/ADP ratio, correlated with sepsis-induced multiple organ failure and poor outcome (Brealey et al, 2002; Fredriksson et al, 2006). This energy imbalance in septic individuals leads to the activation of a starvation response, as evidenced by the following observations. (1) Adipose tissue in septic patients shows increased lipolysis, resulting in higher blood levels of free fatty acids (FFAs), triglycerides, and glycerol, which are significantly higher in septic non-survivors compared to sepsis survivors (Ilias et al, 2014; Rittig et al, 2016; Langley et al, 2013; Lee et al, 2015; Wang et al, 2020). (2) Hepatic cellular glycogen reserves are heavily depleted (Vandewalle et al, 2021). (3) Skeletal muscle proteolysis is activated, leading to increased concentrations of amino acids (e.g., alanine and glutamine) in the blood (Long et al, 1981; Su et al, 2015; Langley et al, 2013; Wang et al, 2020). Interestingly, many studies show that the conversion of these energy-rich substrates into useful metabolites (e.g., acetyl-CoA, ketone bodies, and glucose) is disturbed in septic individuals. A proteomic and metabolomic study on plasma samples of sepsis patients shows decreased protein levels of nine fatty acid transporters but elevated protein levels of two fatty acid-binding proteins in sepsis non-survivors, suggesting a profound defect in fatty acid β-oxidation (Langley et al, 2013, 2014). This defect is caused by impaired functioning of the peroxisome proliferator-activated α receptor (PPARα) as septic individuals exhibit reduced expression of PPARα and PPARα-dependent genes correlating with severity (Wong et al, 2009; Standage et al, 2012; Van Wyngene et al, 2020). This results in harmful FFA accumulation leading to lipotoxicity and contributing to multiple organ damage and failure (Van Wyngene et al, 2020). On the other hand, septic non-survivors show elevated plasma levels of citrate, malate, pyruvate, dihydroxyacetone, lactate, and gluconeogenic amino acids, suggesting alterations in glycolysis, the TCA cycle and gluconeogenesis (GNEO) pathways (Langley et al, 2013, 2014; Wang et al, 2020). Indeed, studies show that GNEO is critical for surviving sepsis but often fails due to several factors. First, the glucocorticoid receptor, which regulates the transcription of genes encoding GNEO enzymes like *Pck1* (encoding PEPCK) and *G6Pc* (encoding G6Pase), becomes dysfunctional in sepsis, leading to impaired GNEO (Vandewalle et al, 2021). Second, the production of pro-inflammatory cytokines following exposure to endotoxic shock decreases the transcription of the rate-limiting enzyme *Pck1* by reducing the expression of the nuclear receptor cofactor PGC1a, the latter being an essential cofactor for *Pck1* transcription (Chichelnitskiy et al, 2009). Lastly, oxidative inhibition of liver G6Pase driven by iron further contributes to GNEO failure during sepsis. Interestingly, counteracting G6Pase repression using ferritin has been shown to preserve GNEO and reduce sepsis mortality, highlighting the importance of sustaining this process during sepsis (Weis et al, 2017). Altogether, this suggests the presence of a sepsis-associated failing starvation response correlating with a bad outcome.

## Pyruvate metabolism in the cytosol

### Pyruvate kinase: the formation of pyruvate

PK catalyzes the final step of glycolysis, involving the irreversible transfer of a phospho-group from PEP to ADP, yielding a molecule of pyruvate and ATP. It is one of the rate-limiting enzymes of glycolysis, alongside hexokinase and phosphofructokinase, and is therefore of crucial importance as a key metabolic control point

(Schormann et al, 2019). Moreover, since PK is responsible for one of the two ATP-producing reactions in glycolysis, it plays a major role in glycolysis-dependent energy production when OXPHOS is insufficient (Gray et al, 2014).

Mammals express four different, tissue-specific PK isozymes encoded by two paralogous genes. The pyruvate kinase type L (PKL) and pyruvate kinase type R (PKR) isozymes are encoded by the same *Pklr* gene, but the expression is driven by different tissue-specific promoters, whereas the pyruvate kinase type M1 (PKM1) and pyruvate kinase type M2 (PKM2) isozymes are generated via alternative splicing by the exclusion of the 10th or 9th exon of the *Pkm* gene, respectively (David et al, 2010; Wong et al, 2014). Each isozyme comprises specific kinetic properties and their expression is tightly regulated to accommodate for the cell-type-specific metabolic needs (Tsutsumi et al, 1988; Prakasam and Bamezai, 2018). For instance, PKL is predominantly found in the liver and to a lesser extend in the kidney and small intestine, while PKR is exclusively found in erythrocytes. PKM1 is expressed in tissues with increased energy needs such as the heart, muscles and brain. Furthermore, PKM2 is generally accepted as the early embryonic form and can be found in high-proliferating cells such as lymphocytes, intestinal epithelial cells, and cancer cells. In addition, the expression pattern of PKM2 changes during development by which it will be progressively replaced by the tissue-specific PK isozymes (Yamada and Noguchi, 1999; Mazurek, 2011).

The PKL, PKR, and PKM1 isozymes are solely active in a homotetrameric form by which each subunit consists of three domains (domains A, B, and C) and a small N-terminal domain. In contrast, the PKM2 isozyme can exist in a more active tetrameric form retaining in the cytosol, but also in a less active dimeric form which can translocate to the nucleus (Muirhead et al, 1986; Dombrauckas et al, 2005). The activity of tetrameric PKM2 is mainly controlled via allosteric regulation. For example, fructose 1,6-bisphosphate (FBP), an upstream glycolytic intermediate, functions as a positive allosteric effector for PK. When FBP is present, it binds to an allosteric effector binding site on domain C resulting in the stabilization of the tetrameric form which increases PEP affinity and consequently PK enzymatic activity (Ashizawa et al, 1991; Jurica et al, 1998). This FBP-dependent regulation applies to PKR, PKL and PKM2 (Ikeda et al, 1997). Conversely, PKM1 is non-allosterically regulated and is therefore constitutively active in the tetrameric form showing the highest PEP affinity (Jurica et al, 1998; Dombrauckas et al, 2005). PK activity can also be allosterically regulated by serine, ATP and phenylalanine, and at the transcriptional, posttranscriptional and posttranslational level (Yamada and Noguchi, 1999).

PK plays a pivotal role in controlling glycolytic metabolic flux. Moreover, numerous studies provide evidence that PKM2 is a crucial mediator of the Warburg effect, i.e., an increased glycolytic rate in the presence of oxygen or also called aerobic glycolysis (Yang and Lu 2013; Warburg, 1956; Luo and Semenza, 2012). In sepsis, metabolic reprogramming from OXPHOS to aerobic glycolysis is a typical hallmark of activated immune cells in order to accommodate their high energy requirements for performing their inflammatory responses. Indeed, serum PKM2 levels are increased in sepsis patients and are positively correlated with blood glucose, lactate, LDH and disease severity (Wang et al, 2023). In addition, urinary PKM2 expression is elevated in patients with sepsis-associated acute kidney injury and positively correlates with

---

**Box 2    The versatile role of lactate in sepsis**

For decades, lactate was considered as an inert by-product of glycolysis rather than a bioactive molecule. Interestingly, lactate can also function as a signaling molecule and can be transported into cells by the monocarboxylate transporters (MCTs), which play a crucial role in metabolic communication between cells. MCT4 is specialized for lactate export, while MCT1 can either import or export lactate, depending on whether it is expressed by oxidative or glycolytic cells.

Lactate is often considered harmful in sepsis; Injection in septic animals worsens sepsis, whereas inhibiting lactate production with the LDH inhibitor, oxamate, improves sepsis survival (Yang et al, 2022; Fei et al, 2024). In macrophages, extracellular lactate uptake leads to HMGB1 lactylation, promoting its release and increasing endothelial permeability (Yang et al, 2022). Inhibiting lactate uptake with CHC, an MCT inhibitor, suppresses lactate-induced HMGB1 lactylation in macrophages. In neutrophils, lactate uptake through MCT1 upregulates PD-L1 expression, delaying apoptosis in these cells. Administration of the selective MCT1 inhibitor AZD3965 increases neutrophil apoptosis leading to enhanced survival in CLP mice (Fei et al, 2024). Research by Vandewalle et al even demonstrates that lactate significantly contributes to sepsis lethality as hyperlactatemia, in combination with a sepsis-induced glucocorticoid resistance, results in a lethal vascular collapse via uncontrolled vascular endothelial barrier dysfunction (Vandewalle et al, 2021).

Contrary to these findings, other researchers have reported that lactate infusion can improve sepsis outcomes by enhancing hemodynamics and reducing inflammation (Walenta et al, 2000). Lactate's role in immunosuppression is well-documented in cancer research and also in inflammatory conditions, where it has been shown to reduce organ damage (Walenta et al, 2000; Hoque et al, 2014). Recently, lactate produced by monocyte-derived human tolerogenic dendritic cells has been found to decrease T-cell proliferation, delaying graft–versus–host disease (Marin et al, 2019).

Lactate also has direct protective effects on organs such as the heart and brain. In the heart, it serves as an energy source, and pharmacological inhibition of MCT4—which blocks lactate export—mitigates heart failure in mice (Cluntun et al, 2021b). Systemic lactate deprivation using dichloroacetate and ICI-118551 (inhibiting respectively PDK and β2-adrenergic receptors) is linked to decreased myocardial energetics, reduced cardiovascular performance, and early death in endotoxic shock (Levy et al, 2007). In the brain, lactate compensates for reduced glucose

uptake following traumatic brain injury (Glenn et al, 2015). Given the starvation response observed in sepsis patients, it stands to reason that lactate is necessary for maintaining myocardial and cerebral metabolism during sepsis (Vandewalle and Libert, 2022).

In addition, lactate can directly affect pathogens. For example, Candida albicans utilizes lactate to increase its resistance to host-relevant stressors, enhance biofilm formation, and evade macrophage recognition (Williams and Lorenz). Conversely, lactate inhibits the growth of Salmonella in vitro and reduces mortality from Salmonella infection in vivo (Iraporda et al, 2017).

In summary, lactate acts as a double-edged sword, with its effects varying based on dose, type (lactic acid versus lactate anion), metabolic state, cell type, and pathology. While high lactate levels are clearly detrimental in sepsis, some clinicians now recognize that mild lactate production during sepsis is a compensatory mechanism. The so-called "Lactate Shuttle Theory" leverages lactate as a signaling and energy molecule to meet increased energy demands and dampen the inflammatory response during sepsis (Brooks, 2018).

Of note, sepsis is frequently associated with lactic acidosis, which is typically defined as a blood lactate concentration exceeding 5 mM combined with a blood pH below 7.35 (Mizock and Falk, 1992). The precise role of glycolysis in inducing lactatemia or lactic acidosis remains unclear. During glycolysis, one molecule of glucose yields two molecules of pyruvate, 2 ATP, 2 NADH, and 2 protons ($H^+$). When pyruvate is converted to lactate by LDH, NADH is reverted to $NAD^+$, consuming an equivalent amount of protons. Thus, an increase in lactate production leading to hyperlactatemia is not inherently acidifying. The primary source of protons, and hence acid, is ATP hydrolysis. The Krebs cycle actively consumes these protons, so a reduction in Krebs cycle flux due to impaired oxygen utilization or mitochondrial dysfunction leads to acid accumulation. Simultaneously, reduced Krebs cycle activity results in lactate production. The cotransport of lactate and protons by MCT causes tissue hypoxia-related acidosis to manifest as "lactic acidosis" clinically. Therefore, the term "lactic acidosis" is a misnomer, and "lactate-associated acidosis" is more accurate, as lactate itself does not cause acidosis (Müller et al, 2023). In sepsis, this lactate-associated acidosis is primarily compensated for by the kidneys, which decrease strong anions, thus widening the strong ion difference. Consequently, the severity of acidemia heavily depends on renal function (Gattinoni et al, 2019).

---

serum creatinine levels (Jiajun et al, 2024). Hence, this suggests that septic patients are characterized by a PKM2-mediated glucose metabolic reprogramming, and that PKM2 could potentially be used as a prognostic biomarker.

Generally, PKM2 can be induced by hypoxia or after LPS stimulation (Wasyluk and Zwolak, 2021). More specifically, LPS-stimulated macrophages show increased PKM2 expression and simultaneous phosphorylation of PKM2 on Tyrosine 105 keeping PKM2 in its dimeric, inactive form. With this, PKM2 can translocate to the nucleus resulting in the activation of hypoxia-inducible factor 1α (HIF1α) via the formation of a PKM2-HIF1α complex (Palsson-Mcdermott et al, 2015). On the one hand, PKM2-mediated activation of HIF1α leads to a metabolic shift towards aerobic glycolysis by inducing the expression of glycolysis-related genes, resulting in elevated lactate production. In turn, lactate inhibits histone deacetylase activity, resulting in high-mobility Box 1 (HMGB1) hyperacetylation and its subsequent release (Yang et al, 2014). In addition, this PKM2-mediated lactate production induces EIF2AK2 phosphorylation resulting in NLRP3 and AIM2 inflammasome activation and subsequent pro-inflammatory mediator release (IL-1β, IL-18, and HMGB1) in LPS-treated macrophages (Xie et al, 2016). On the other hand,

PKM2-HIF1α complex can bind to the IL-1β promoter and induces excessive IL-1β production in LPS-activated macrophages (Palsson-Mcdermott et al, 2015). Furthermore, these studies indicate that the potential PKM2 inhibitor, shikonin, reduces PKM2 activity in LPS-stimulated macrophages. This reduction protects mice from lethal endotoxemia and sepsis by partially decreasing lactate production, and consequently pro-inflammatory cytokines release (Yang et al, 2014; Xie et al, 2016). This study confirms the need for therapeutic interventions addressing PKM2-mediated aerobic glycolysis as a metabolic control in inflammation for the treatment of sepsis. Indeed, celastrol, a natural anti-inflammatory compound, provides protection against lethal endotoxemia and sepsis in mice by reducing tissue damage. This protection mechanism operates through the binding of celastrol to Cys424 of PKM2, inhibiting its enzymatic activity and consequently alleviating the Warburg effect and the secretion of pro-inflammatory cytokines in LPS-stimulated macrophages (Luo et al, 2022). Notably, sepsis is characterized by sepsis-induced cardiomyopathy (SIC) which is correlated with high mortality (Court et al, 2002). Ni et al demonstrate that LPS-stimulated cardiomyocytes are characterized by increased PKM2 expression and that this is necessary for protection against myocardial injury (Ni et al, 2022). This

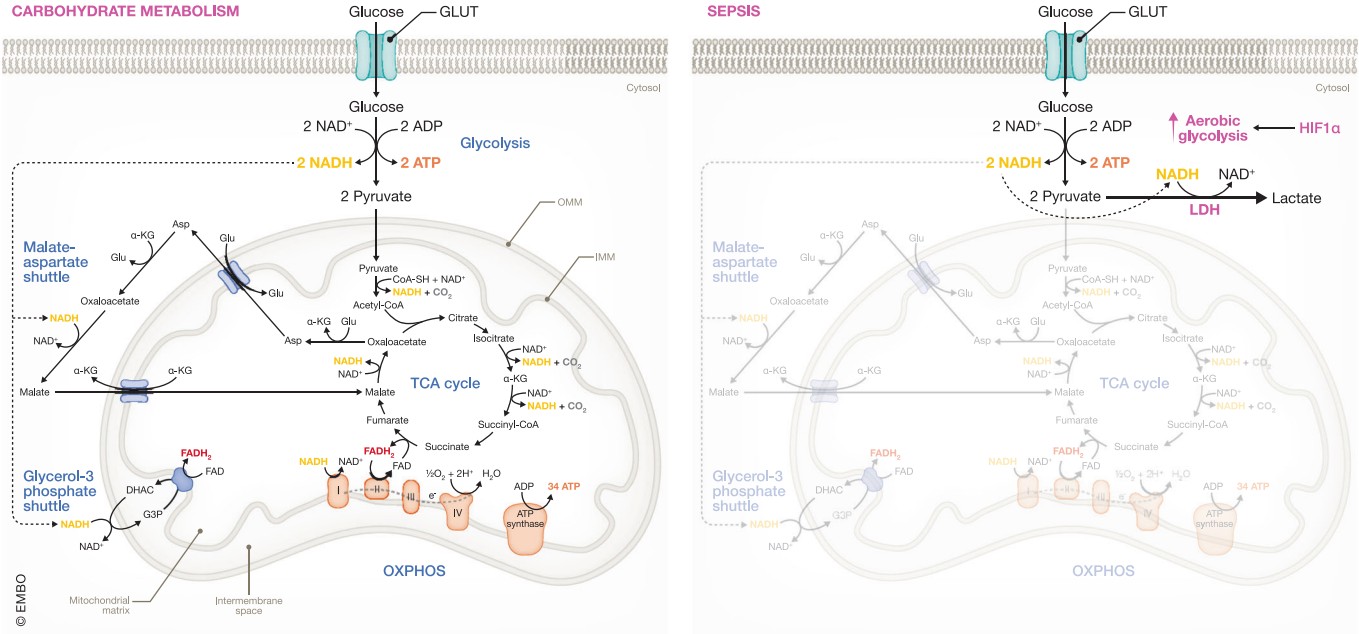

**Figure 1. Overview of carbohydrate metabolism and its alterations in sepsis.**

In physiological conditions, glucose is imported in the cells via glucose transporters (GLUT) where it is converted into two pyruvate molecules with the additional formation of 2 NADH and 2 ATP molecules in the catabolic process called glycolysis. When oxygen levels are sufficient, pyruvate is imported in the mitochondria and is oxidized by pyruvate dehydrogenase complex to generate acetyl-CoA which enters the tricarboxylic acid (TCA) cycle to generate reducing equivalents, NADH and $FADH_2$, to drive oxidative phosphorylation (OXPHOS)-dependent ATP production. In addition, the formed cytosolic NADH will be transferred into the mitochondria by two shuttle systems, i.e., malate-aspartate shuttle and glycerol-3-phosphate shuttle, thereby regenerating $NAD^+$. Upon sepsis, cytopathic hypoxia results in a metabolic shift from OXPHOS towards aerobic glycolysis as a primary, but less efficient source for ATP production which is mediated by hypoxia-inducible factor 1 α (HIF1α). The formed cytosolic pyruvate is predominantly converted into lactate by lactate dehydrogenase (LDH) thereby regenerating $NAD^+$ from NADH for maintaining glycolysis activity. ADP adenosine diphosphate, ATP adenosine triphosphate, IMM inner mitochondrial membrane, OMM outer mitochondrial membrane, NAD(H) nicotinamide adenine dinucleotide (+ hydrogen), $FADH_2$ flavin adenine dinucleotide, Glu glutamate, α-KG α-ketoglutarate, DHAC dihydroxyacetone phosphate, G3P glycerol-3-phosphate. Adapted from Van Wyngene et al, 2018.

protection is mediated by the interaction of PKM2 with sarcoplasmic/endoplasmic reticulum calcium ATPase 2a (SER-CA2a), thereby maintaining cardiac calcium homeostasis and cardiac contraction. PKM2-mediated protection against SIC is also necessary to attenuate LPS-induced mitochondrial damage marked by diminished ATP production, reduced mitochondrial respiratory complex I/III activities and increased ROS production, eventually resulting in enhanced myocardial inflammation and impaired cardiac function. This is mediated via PKM2-dependent phosphorylation of prohibitin 2 (essential for the preservation of mitochondrial function and structure) in order to limit LPS-mediated prohibitin 2 degradation (Ren et al, 2024; Du et al, 2024). Hence, cardiomyocyte-specific PKM2 overexpression exhibits protective effects against SIC in mice with LPS or Gram-negative bacteria-induced sepsis (Ni et al, 2022; Du et al, 2024).

In conclusion, both PKM2 inhibition and activation can improve sepsis survival in a cell-type-dependent way. The PKM2 inhibitors, shikonin and celastrol, mainly reduce the Warburg phenotype in activated macrophages thereby reducing lactate production and subsequent pro-inflammatory responses while cardiomyocyte-specific PKM2 overexpression improves myocardial injury during SIC in septic subjects. In view of this, it seems possible that the administration of PKM2 inhibitors negatively affects SIC. Therefore, further research is needed to optimize PKM2

therapy in an organ-specific way. It is suggested to specifically target PKM2 inhibitors towards highly aerobic glycolytic tissues to reduce lactate production and to target PKM2 activators towards the heart to reduce SIC.

### Lactate dehydrogenase

LDH is a cytosolic enzyme that catalyzes the reduction of pyruvate into lactate with the concomitant oxidation of NADH into $NAD^+$ and vice versa. This reaction is essential in anaerobic metabolism since low oxygen levels result in reduced OXPHOS capacity. Hence, energy production is solely dependent on anaerobic glycolysis where LDH activity is responsible for the regeneration of $NAD^+$ to drive glycolysis-dependent ATP production (Khan et al, 2020). Nevertheless, lactate production during the anaerobic breakdown of glucose in the skeletal muscle is a metabolic dead end. Therefore, the formed lactate is released into the bloodstream and transported to the liver where it can be converted into pyruvate via LDH as part of the Cori cycle. In this Cori cycle in hepatocytes, the produced pyruvate will be transformed into glucose via GNEO, serving as a source of energy for the skeletal muscle (Passarella, 2018) (Fig. 3).

LDH is a tetrameric enzyme belonging to the class of oxidoreductases and is ubiquitously expressed in all body tissues. The LDH tetrameric structure comprises two subunit types, namely the skeletal muscle (M) type, encoded by the *Ldha* gene, and the

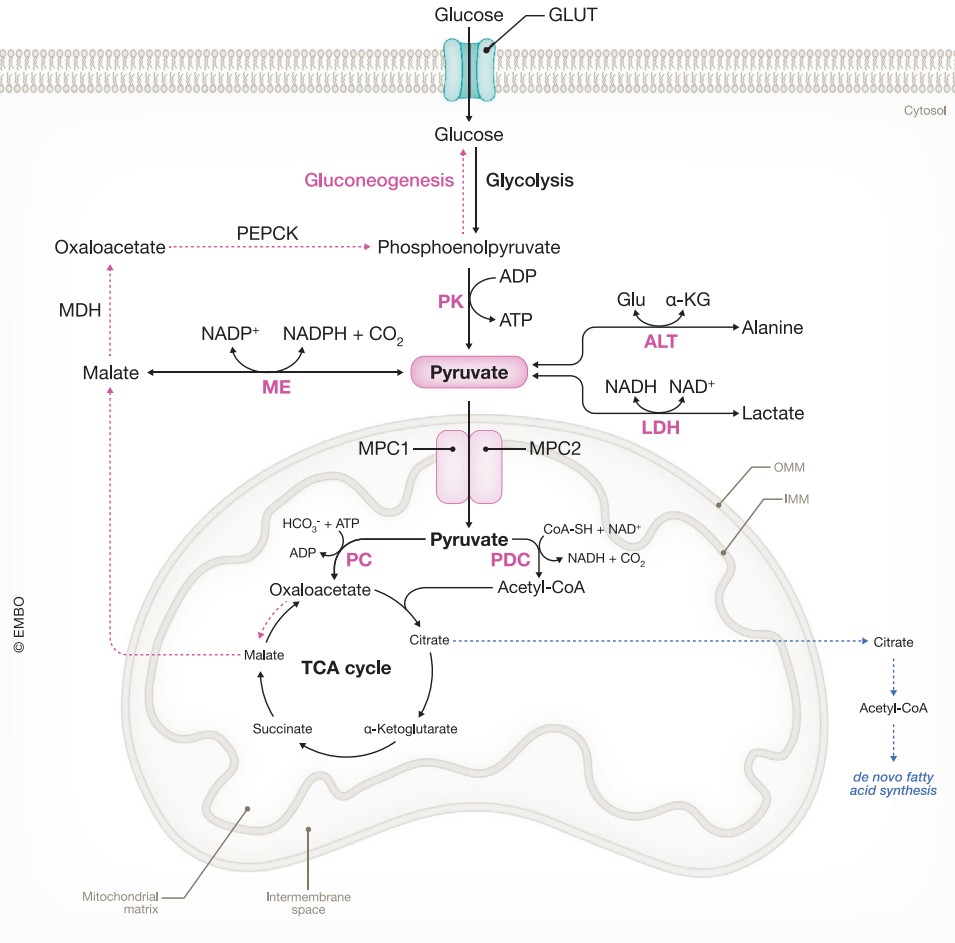

**Figure 2. The metabolic fates of pyruvate.**

In normal eukaryotic cells, glycolysis primarily produces pyruvate, catalyzed by pyruvate kinase (PK) converting phosphoenolpyruvate into pyruvate. Cytosolic pyruvate can also be formed via the oxidative decarboxylation of malate to pyruvate and $CO_2$ by malic enzyme (ME). A small portion of the cytosolic pyruvate can be reduced to lactate by lactate dehydrogenase (LDH) or transaminated to alanine by alanine aminotransferase (ALT). However, the main route of cytosolic pyruvate is transportation across the inner mitochondrial membrane (IMM) by the mitochondrial pyruvate carrier (MPC) into the mitochondrial matrix. Depending on the cellular energy needs, transported pyruvate can be oxidized by pyruvate dehydrogenase complex (PDC) to produce $CO_2$ and acetyl-CoA, which enters the tricarboxylic acid (TCA) cycle to generate energy or can be used for de novo fatty acid synthesis. Alternatively, mitochondrial pyruvate can be carboxylated by pyruvate carboxylase (PC) to generate oxaloacetate, which can fuel the TCA cycle or contribute to pyruvate-driven gluconeogenesis. GLUT glucose transporter, MDH malate dehydrogenase, PEPCK phosphoenolpyruvate carboxykinase, ADP adenosine diphosphate, ATP adenosine triphosphate, NAD(P)(H) nicotinamide adenine dinucleotide (phosphate) ( + hydrogen), CoA coenzyme A, $HCO_3^-$ bicarbonate, OMM outer mitochondrial membrane.

heart (H) type, encoded by the *Ldhb* gene. The subunits are called M and H as these are primarily found in skeletal muscle and the heart, respectively (Stambaugh and Post, 1966). Five different LDH isozymes have been identified, each composed with specific proportions of M and H subunits: LDH-1 ($H_4$), LDH-2 ($H_3,M_1$), LDH-3 ($H_2,M_2$), LDH-4 ($H_1, M_3$) and LDH-5 ($M_4$) (Kopperschläger and Kirchberger, 1996; Read et al, 2001). The LDH isozymes perform the same biochemical reaction, but its occurrence is organ-specific and depends on the energetic status of the respective tissue. For instance, LDH-1 is predominantly found in the heart, brain and red blood cells (RBCs), LDH-2 in white blood cells, LDH-3 in the lungs, LDH-4 in the kidneys, pancreas, and placenta, and LDH-5 in the liver and skeletal muscle (Read et al, 2001). Moreover, the M subunit has a higher affinity for pyruvate thereby promoting the formation of lactate while the H subunit has

a higher affinity for lactate thereby promoting the formation of pyruvate (Kopperschläger and Kirchberger, 1996). This corresponds with the metabolic profile of the respective tissue as the skeletal muscle is mainly known as a highly glycolytic tissue resulting in lactate production. In contrast, the heart primarily relies on lactate oxidation to generate energy.

LDH activity is controlled at three main levels, transcriptional, posttranscriptional and substrate-level regulation. Firstly, the LDHA promoter region comprises consensus sequences for key transcription factors like HIF1α, c-Myc, forkhead box protein M1 (FOXM1), and Kruppel-like factor 4 (KLF4) (Firth et al, 1995; Lewis et al, 1997; Cui et al, 2014; Shi et al, 2014). Secondly, it has been demonstrated that LDHA phosphorylation by the oncogenic receptor tyrosine kinase FGFR1 induces the formation of the active, tetrameric form and NADH substrate binding (Fan et al, 2011). In addition, deacetylation

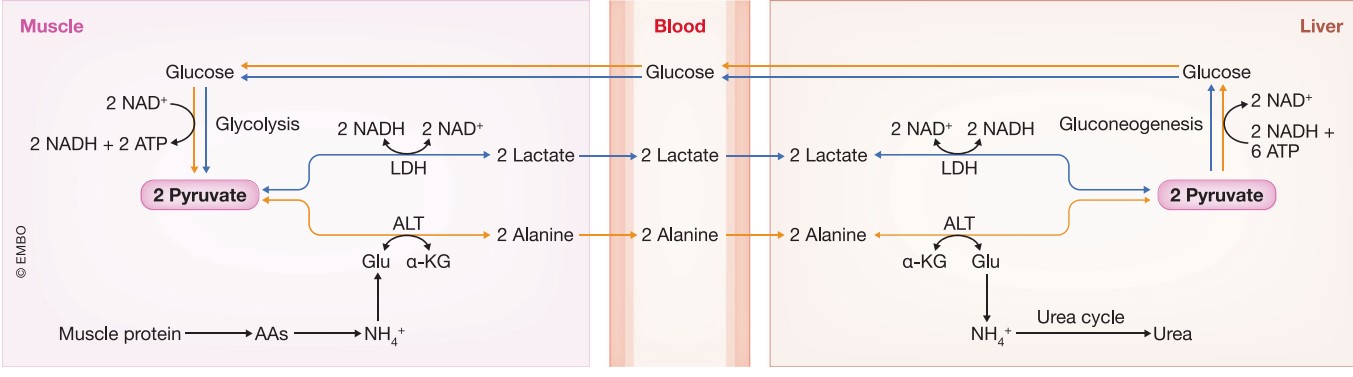

**Figure 3. Overview of the Cori & the Cahill cycle.**

**Cori cycle**: During periods of heightened muscular activity, oxygen availability becomes limited for adequate energy production through aerobic respiration. This results in the breakdown of glucose via anaerobic glycolysis, leading to the conversion of pyruvate into lactate by lactate dehydrogenase (LDH). The formed lactate is then released into the bloodstream and transported to the liver. Within the liver, gluconeogenesis occurs, where lactate is initially converted back to pyruvate by LDH which is subsequently converted back to glucose. In turn, the formed glucose is again utilized by skeletal muscle as a fuel source, completing the cycle by being metabolized back into lactate. **Cahill cycle**: Proteolysis in the skeletal muscle generates amino acids (AAs) for energy. The formed ammonia ($NH_4^+$) is transaminated to pyruvate to form alanine by alanine aminotransferase (ALT). Alanine diffuses into the bloodstream and is exported to the liver. In the liver, the amino group of alanine is transferred to alpha-ketoglutarate (α-KG) by ALT, resulting in the formation of pyruvate and glutamate (Glu), respectively. The formed glutamate is deaminated resulting in the release of its amino group as ammonia, which is then eliminated via the urea cycle. The generated pyruvate is metabolized by gluconeogenesis into glucose which will be transported back to the skeletal muscle where it will be used by glycolysis for energy production. ATP adenosine triphosphate, NAD(H) nicotinamide adenine dinucleotide (+ hydrogen).

of LDHA by SIRT2 deacetylase increases LDHA activity in pancreatic cancer (Zhao et al, 2013). Finally, LDH activity is also modulated by the relative availability and concentration of its substrates. For instance, extreme muscular activity is accompanied by an increased glycolytic flux due to limited oxygen availability for OXPHOS, leading to excessive production of pyruvate and NADH. However, this exceeds the metabolism capacity of PDC and NADH shuttle enzymes resulting in the flux of pyruvate and NADH through LDH (Spriet et al, 2000; Valvona et al, 2016).

As mentioned in the previous section, sepsis is associated with glucose metabolic reprogramming of immune cells (i.e., Warburg phenotype) potentially mediated via elevated HIF1α activation. Apart from PKM2 and hypoxic conditions, bacterial products (e.g., LPS via mTOR activation) and pro-inflammatory cytokines (e.g., TNFα) can induce HIF1α activity under normoxic conditions (Nishi et al, 2008; Regueira et al, 2009). In turn, HIF1α induces the expression of glycolysis-related genes including LDH, thereby increasing the aerobic glycolytic flux responsible for elevated pyruvate levels driving LDH activity in producing lactate (Garcia-Alvarez et al, 2014; Tan et al, 2021). Indeed, several clinical studies have provided evidence that serum lactate levels are elevated and may hold significance in predicting mortality in patients with septic shock (defined by Sepsis-3) (Ryoo et al, 2018; Lee et al, 2021). Moreover, sepsis patients show increased serum LDH levels which is positively correlated with serum lactate, IL-1β and 28-day mortality, thereby suggesting that glucose metabolic reprogramming of immune cells contributes to sepsis mortality (Lu et al, 2018; Frenkel et al, 2023). Moreover, many in vitro studies show increased levels of LDHA upon LPS stimulation in macrophages (Yang et al, 2014; Palsson-Mcdermott et al, 2015; Xie et al, 2016; Zhang et al, 2022). Apart from lactate production by immune cells,

Levy et al observe elevated levels of pyruvate and lactate in the skeletal muscle of septic shock patients. Interestingly, this is driven by the release of ephinephrine which triggers $Na^+K^+$ ATPase activity to induce membrane hyperpolarization. This increased activity of the $Na^+K^+$ pump necessitates greater ATP consumption, which is facilitated by an enhanced aerobic glycolysis potential, and thus increased LDH activity. Consequently, this process is associated with $Na^+K^+$ ATPase-linked lactate production in skeletal muscle, as evidenced by the administration of a specific $Na^+K^+$ ATPase inhibitor, ouabain, which significantly reduces skeletal muscle pyruvate and lactate levels in patients with septic shock (Levy et al, 2005). Altogether, this highlights the need for therapeutic interventions targeting LDHA in sepsis.

Interestingly, the specific LDH inhibitor, sodium oxalate, has shown promise in enhancing survival rates in polymicrobial sepsis by decreasing lactate production. This protective effect results from the role of lactate in mediating HMGB1 lactylation via its direct uptake through monocarboxylate transporters (MCTs) or via a p300/CBP-dependent mechanism in activated macrophages. Lactate can also bind to its receptor, G-protein coupled receptor 81 (GPR81), leading to decreased expression of SIRT1 deacetylase, thereby inducing increased HMGB1 acetylation in activated macrophages. Eventually, lactylation and acetylation of HMGB1 lead to its exosomal secretion in macrophages, contributing to endothelial barrier dysfunction and vascular permeability which exacerbates sepsis progression (Yang et al, 2022). Zhang et al have discovered that capsaicin effectively inhibits LDHA activity and hence, the Warburg effect in LPS-stimulated macrophages and LPS-induced endotoxemia in mice. In this way, capsaicin treatment could alleviate the inflammatory storm resulting in reduced multiple organ failure in lethal endotoxemia and sepsis (Zhang

et al, 2022). As previously mentioned, celastrol mitigates sepsis survival by reducing pro-inflammatory cytokine release via PKM2 inhibition. Strikingly, this inhibition extends beyond PKM2, as celastrol also targets LDHA activity (Luo et al, 2022). Conversely, it should be noted that LDHA activity could also have beneficial effects. For instance, neutrophils typically have a vital function in the frontline defense of the innate immune system by eliminating pathogenic bacteria at the site of infection. However, in septic patients, neutrophil dysfunction is apparent as septic polymorpho-nuclear neutrophils (PMNs) exhibit reduced chemotaxis and phagocytosis capacities compared to PMNs from patients with a non-septic infection (Pan et al, 2022). Strikingly, Pan et al have revealed that this comprised immune function of septic PMNs is linked with a diminished Warburg phenotype and downregulation of LDHA, normally induced via the PI3K/Akt/HIF1α pathway. These findings are corroborated by administering glycolysis, PI3K/Akt or HIF1α inhibitors to LPS-tolerant neutrophils (i.e., septic PMNs) resulting in reduced lactate production, neutrophil chemotaxis, and phagocytosis. Hence, this underscores the importance of the activation of the LDHA-mediated Warburg phenotype in regulating phagocytosis of septic PMNs in vitro and in vivo to combat the infection (Pan et al, 2022).

In summary, there is a clear discrepancy in therapeutically targeting the LDHA-mediated aerobic glycolysis in activated immune cells and other tissues. While inhibiting LDHA activity primarily mitigates sepsis progression and improves survival by diminishing lactate production and, consequently, lactate-mediated toxic effects, e.g., HMGB1 release by macrophages, it also hinders the Warburg phenotype in macrophages and neutrophils, thereby limiting excessive inflammatory responses. However, LDH inhibition may exacerbate neutrophilic dysfunctions, reducing bacterial clearance during sepsis. Hence, it would make sense to only inhibit LDH in highly glycolytic tissues, such as skeletal muscles. Moreover, further research is warranted to prioritize therapies that counteract lactate's toxic effects via GPR81 signaling, e.g., GPR81 inhibition by 3-hydroxy-butyrate, rather than solely inhibiting lactate production (Shen et al, 2015). In this way, LDH can continue to perform its necessary functions, particularly in septic neutrophils.

### Alanine aminotransferase

ALT, also known as glutamic pyruvate transaminase (GPT), catalyzes the reversible transamination between alanine and α-ketoglutarate to form pyruvate and glutamate. With this, ALT plays an important role in linking amino acid and carbohydrate metabolism which is mediated by the alanine cycle or Cahill cycle (Felig, 1973; Gray et al, 2014). The cycle starts in the muscle where pyruvate, generated from intramyocellular glycolysis, is transaminated with ammonia, generated from muscle protein catabolism, to form alanine which is exported to the liver. In the liver, alanine is transaminated back by ALT to pyruvate which can either be used to drive the TCA cycle flux or it can be processed by GNEO to produce glucose. Eventually, glucose is transported back to the skeletal muscle where it will be consumed by glycolysis for energy production (Felig, 1973). The alanine cycle shows close similarity with the Cori cycle, but it is less efficient as the deamination of glutamate releases its amino group as ammonia, which needs to be detoxified via the urea cycle (Fig. 3) (Brosnan, 2000).

Two isoforms of ALT, ALT1, and ALT2, have been recognized and are encoded by two different genes annotated as *Gpt* and *Gpt2*,

respectively. These isoforms exhibit differences in their cellular location as ALT1 is a cytosolic protein while ALT2 is found in the mitochondrial matrix. In addition, ALT1 and ALT2 display variations in their tissue distribution. ALT1 expression is generally present in human liver, skeletal muscle, myocardium, and kidney. ALT2 is primarily found in the skeletal muscle and the heart (Lindblom et al, 2007; Yang et al, 2009). In addition, there is significant variation in the expression of ALT1 and ALT2 across different species. For example, when compared to their distribution in humans, both ALT1 and ALT2 are notably more highly expressed in the livers of rats and mice (Rafter et al, 2012). However, little is known about the regulation of ALT activity. In general, pyridoxal phosphate is required as a coenzyme for ALT activity, functioning as an amino carrier (Cheung, 1975). In rat liver, it has been reported that ALT expression can be induced by factors such as high protein intake, fasting, cortisol, glucagon and (nor)epinephrine (Rosen et al, 1959; Coss et al, 2012). In addition, there is evidence suggesting that the expression of ALT2 is regulated by androgens through activation of one or more promoter androgen response elements (Begum and Datta, 1992).

According to literature, limited research have been conducted towards ALT activity and its significance in sepsis. Nevertheless, some studies indicate that alanine-driven GNEO is disturbed during sepsis. Indeed, a metabolomics analysis of plasma samples collected from septic shock patients with a Sequential Organ Failure Assessment score >8 has revealed higher alanine levels in patients non-responsive to supportive care. This postulates hepatic problems with the glucose–alanine cycle (i.e., alanine-driven GNEO) or with the GNEO pathway in general, as indicated by concurrent increases in pyruvic acid and lactic acid levels at later stages (Cambiaghi et al, 2017). Supporting this, Galia et al have demonstrated that GNEO from precursors alanine, lactate and pyruvate is significantly reduced in septic rats (de Souza Galia et al, 2021). Hence, it seems clear that the failure in the hepatic glucose–alanine cycling is due to general GNEO problems in septic subjects. However, further research is needed to pinpoint if these issues with alanine processing already occur at the ALT level or if this is just a broader issue in the GNEO pathway. Of note, serum ALT levels are extensively used as a clinical biomarker of hepatic health, given that ALT is highly expressed in liver tissue. With this, serum of septic patients is typically characterized by increased ALT levels due to liver dysfunction (Gu et al, 2018).

Interestingly, it noteworthy that many other studies have already highlighted the importance of alanine metabolism, particularly ALT activity, in adjusting to changes in hepatic metabolic states such as starvation and obesity. Since sepsis is characterized by a compromised starvation response, these studies should be taken into consideration for further exploration of the role of ALT activity in hepatic metabolism during sepsis (Vandewalle and Libert, 2022). For instance, Petersen et al demonstrate a starvation-induced reduction in glucose–alanine cycling, which in turn reduces the hepatic PC flux and leads to a decrease in endogenous glucose production and hepatic mitochondrial oxidation. These findings underscore a potential role of glucose–alanine cycling in regulating hepatic OXPHOS (Petersen et al, 2019). In line with this, Okun et al show increased hepatic ALT levels in mice with type 2 diabetes (T2D) and obesity which is associated with decreased serum alanine levels and increased blood glucose levels. Interestingly, liver-specific ALT silencing hampers hyperglycemia and

improves skeletal muscle size and function. This indicates that alanine catabolism by ALT plays a crucial role in glycemic regulation and muscle atrophy in T2D and obesity (Okun et al, 2021). These findings align with another recent investigation by Martino et al who states that ALT2 levels are elevated in the liver of diet-induced obese mice which is mediated by the ER stress-activated transcription factor ATF4. Furthermore, liver-specific deletion of ALT2 in obese mice reduces blood glucose levels and amino acid dependent GNEO (Martino et al, 2022).

Overall, these data illustrate a pivotal role of ALT activity in regulating blood glucose levels in mice with altered hepatic metabolism. Remarkably, the most severe sepsis patients are characterized by hypoglycemia, which correlates with increased mortality (Mitsuyama et al, 2022). It is speculated that these reduced blood glucose levels may be attributed to decreased ALT activity in the livers of septic individuals. Therefore, investigating the effects of hepatocyte-specific ALT overexpression or silencing on sepsis blood glucose levels and outcomes could be of interest. In addition, given the intriguing role of glucose–alanine cycling in controlling hepatic OXPHOS during starvation, exploring whether stimulating ALT activity could alleviate mitochondrial OXPHOS problems observed during sepsis warrants further investigation. Finally, it should be noted that there is a paradoxically relation between obesity and survival, i.e., sepsis outcomes for overweight and obese patients are more favorable compared to those for individuals with normal weight (Ng and Eikermann, 2017). Hence, it is suggested to explore whether the decreased sepsis sensitivity observed in obese mice is due to enhanced ALT activity leading to elevated alanine-driven GNEO and diminished hypoglycemia in sepsis.

## Pyruvate metabolism in the mitochondria

### Mitochondrial pyruvate import

Pyruvate transport into mitochondria is essential for linking cytosolic and mitochondrial energy metabolism, and oxidizing pyruvate by $O_2$ to yield ATP, water and $CO_2$. Pyruvate can diffuse through the outer mitochondrial membrane (OMM) via porins. However, the passage through the inner mitochondrial membrane (IMM) into the matrix relies on a dedicated carrier known as MPC. The recently discovered MPC is a hetero-oligomeric complex consisting of two proteins, MPC1 (12 kDa, encoded by *Mpc1*) and MPC2 (14 kDa, encoded by *Mpc2*), located in the IMM. While both subunits are ubiquitously expressed, they are notably abundant in certain tissues, including, the liver, kidney, heart, muscles, brown adipose tissue, and brain. These organs have a significant metabolic demand for mitochondrial pyruvate metabolism (Bowman et al, 2016). Little is known about the regulation and working mechanism of this carrier, but both proteins are necessary to maintain MPC complex activity and stability since loss of one protein results in the degradation of the other, leading to a destabilized, inactive MPC complex (Bricker et al, 2012; Herzig et al, 2012). It is also suggested that mitochondrial pyruvate uptake is linked to the electrochemical gradient, involving the simultaneous import of one proton or with the antiport of one hydroxide ion (Halestrap, 1978).

While there is limited exploration of MPC in bacterial sepsis models, certain studies show an important role of MPC-mediated metabolism in the activation of LPS-stimulated macrophages and LPS-induced endotoxemia. Administration of a small molecule MPC inhibitor, UK5099, inhibits pyruvate import into the mitochondria and results in reduced mitochondrial respiration and diminished pro-inflammatory gene expression in both LPS-treated macrophages and bone marrow-derived macrophages (BMDMs). This indicates that a proper flux of pyruvate into the mitochondria to induce pyruvate oxidation via PDC is necessary to maintain pro-inflammatory activation of macrophages (Meiser et al, 2016; Lauterbach et al, 2019). In addition, Ran et al further elucidate the role of MPC in macrophage activation as previous studies have only focused on pharmacological inhibition of MPC and in vitro experiments. They found a dose-dependent effect of UK5099 on the inflammatory response and metabolic reprogramming of BMDMs. Strikingly, this effect has been found to be independent of MPC. BMDMs derived from a murine MPC conditional KO model are only characterized with a reduced glucose flux into the mitochondria, but it had no influence on metabolic reprogramming, ATP production and the inflammatory response after LPS stimulation. Moreover, mice with a conditional *Mpc* deletion in myeloid cells do not show any differences in inflammatory responses and M1 macrophage polarization in a LPS endotoxemia model (Ran et al, 2023). Overall, these findings suggest that UK5099 suppresses metabolic reprogramming and M1 macrophage activation in an MPC-independent manner, potentially through off-target effects such as HIF1α stabilization, rather than directly inhibiting MPC (Ran et al, 2023). Conversely, a recent study of Zhu et al demonstrates that myeloid-specific *Mpc* deletion or pharmacological inhibition of MPC with MSDC-0602K (MSDC) mitigates disease severity after influenza or SARS-CoV-2-induced pneumonia. Specifically, MSDC administration targets lung macrophages (and not BMDM) leading to a suppressed pulmonary hyperinflammatory response via increased mitochondrial fitness resulting in reduced levels of HIF1α-stabilizing metabolites (such as succinate and/or acetyl-CoA) and hence reduced HIF1α levels. This illustrates that mitochondrial pyruvate import via MPC is essential to induce inflammatory responses of lung macrophages, and not BMDM, upon viral infections such as SARS-CoV-2 (Zhu et al, 2023; Ran et al, 2023).

Notably, other studies have been conducted to investigate the role of MPC activity in metabolism and other metabolic diseases. For instance, hepatic loss of either MPC1 or MPC2 has shown that MPC activity is crucial in gating pyruvate-driven GNEO and TCA cycle flux. This is accompanied with the activation of pyruvate-alanine cycling and with increased glutaminolysis to compensate for the impaired pyruvate-driven GNEO and reduced pyruvate-dependent replenishment of TCA cycle intermediates, respectively (Fig. 4) (Gray et al, 2015; McCommis et al, 2015). As increased GNEO is a hallmark in T2D and as MPC plays a pivotal role in regulating pyruvate-driven GNEO, targeting MPC in T2D could be of interest in reducing hyperglycemia. Indeed, MPC activity is significantly increased in mice with diet-induced obesity (i.e., a model for human T2D) and liver-specific deletion of Mpc1 reduces the development of hyperglycemia and glucose intolerance in this mouse model of T2D (Gray et al, 2015; Rauckhorst et al, 2017). In line with this, Zaprinast and 7ACC2, two small molecules that function as MPC inhibitors, have been shown to have antidiabetic effects by suppressing hepatic GNEO and hence reducing blood glucose levels (Hodges et al, 2022). In addition, other studies show the importance of MPC expression and activity in heart failure and cardiac protection. Failing hearts are characterized by MPC

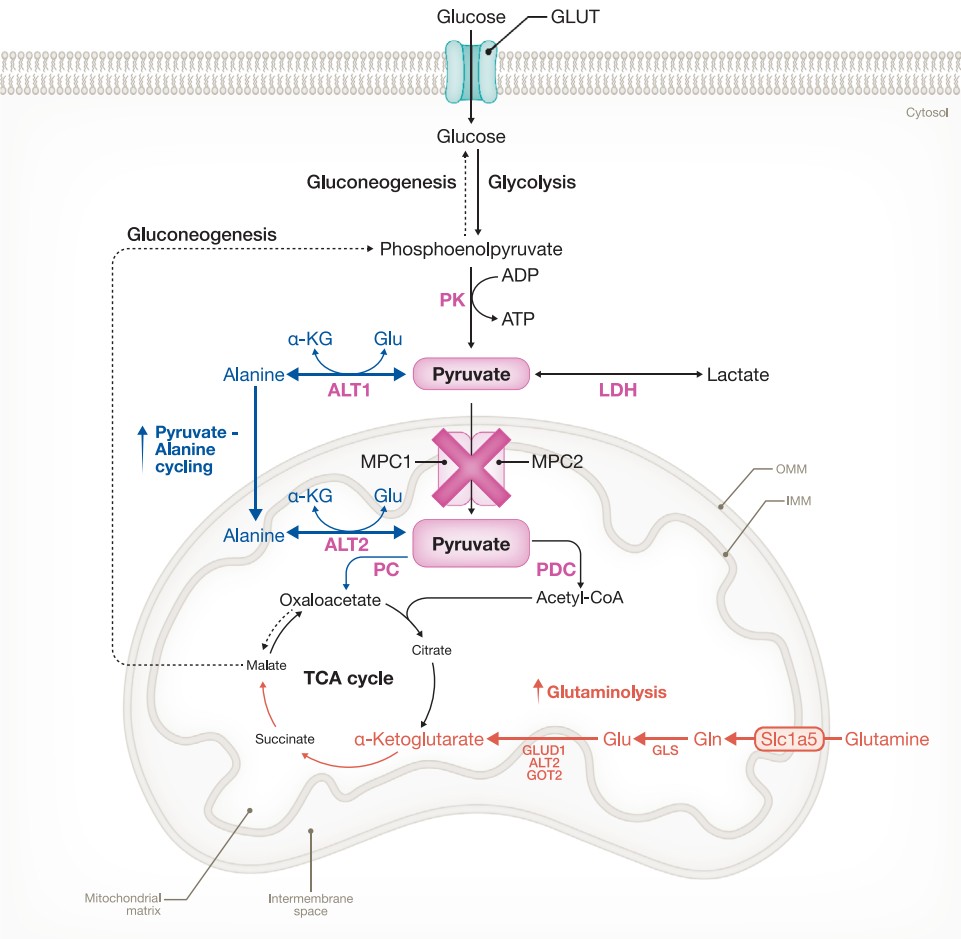

**Figure 4.   Pyruvate-driven gluconeogenesis is compensated by pyruvate-alanine cycling and glutaminolysis during liver-specific loss in MPC activity.**

**Pyruvate-Alanine cycling**: In the absence of Mitochondrial pyruvate carrier (MPC) activity, cytosolic alanine aminotransferase (ALT1) transaminates pyruvate into alanine. The formed alanine is transported into the mitochondrial matrix via an unidentified alanine transporter situated in the inner mitochondrial membrane (IMM). In turn, mitochondrial alanine aminotransferase (ALT2) converts alanine back into pyruvate. The formed mitochondrial pyruvate can then be processed by pyruvate carboxylase (PC) or pyruvate dehydrogenase complex (PDC) to stimulate either pyruvate-driven gluconeogenesis or tricarboxylic acid (TCA) cycle flux, respectively. **Glutaminolysis**: Loss of MPC activity leads to an adaptive usage of glutamine (Gln) for TCA anaplerosis. In this regard, Gln is transported across the IMM into the mitochondrial matrix via a Slc1a5 transporter variant. Within the mitochondrial matrix, Gln is converted into glutamate (Glu) by glutaminase (GLS) and the formed Glu is subsequently deaminated into alpha-ketoglutarate ($\alpha$-KG) by glutamate dehydrogenase (GLUD1) or by two mitochondrial aminotransferases, ALT2 or aspartate aminotransferase (GOT2). The generated $\alpha$-KG can be used to fuel the TCA cycle or support gluconeogenesis. PK pyruvate kinase, LDH lactate dehydrogenase, OMM outer mitochondrial membrane.

downregulation which is responsible for a reduced cardiac glucose/ pyruvate oxidation as cardiac-specific *Mpc1* or *Mpc2* knockout (CS-MPCKO) mice show defects in mitochondrial pyruvate metabolism and in the TCA cycle flux. This metabolic remodeling is further associated with a more anabolic program as the failing hearts of CS-MPCKO mice exhibit elevated levels of anabolic metabolites, including pentose phosphate pathway intermediates and amino acids. Interestingly, the onset of cardiomyopathy in CS-MPCKO mice can be completely hindered by a high-fat, low-carbohydrate ketogenic diet due to stimulation of fatty acid oxidation on the one hand, and by inducible cardiomyocyte MPC1/2 overexpression on the other (McCommis et al, 2020; Fernandez-Caggiano et al, 2020; Cluntun et al, 2021a).

Overall, the role of MPC in sepsis could be twofold. On the one hand, it seems that MPC-mediated pyruvate translocation in the mitochondria is important for inducing macrophage inflammatory responses. However, it should be noted that this effect is dependent on the type of infection (bacterial vs viral) and the specific cell type (BMDM vs lung macrophages). On the other hand, MPC activity could potentially contribute to metabolic dysregulations during sepsis, but this warrants further investigation. With this, studies have demonstrated that MPC plays a pivotal role in pyruvate-driven GNEO and TCA cycle rate. Given that septic individuals exhibit disrupted hepatic GNEO, hypoglycemia, and reduced OXPHOS, it is plausible that a potential loss of MPC activity underlies these phenomena. Therefore, examining the effects of administering UK5099 or hepatocyte-specific MPC1 or MPC2 KO on sepsis survival and pyruvate-mediated GNEO and OXPHOS could be insightful. In addition, SIC is a typical phenomenon in septic individuals. Hence, it is suggested that CS-MPCKO mice exacerbate this phenotype, which is associated with worsened defects in cardiac mitochondrial pyruvate metabolism and TCA

cycle flux. Finally, if MPC activity indeed plays a crucial role in gating hepatic GNEO and SIC in septic subjects, improving survival could be achieved by overexpressing or enhancing MPC activity. Therefore, characterization of the cause of this potential loss in MPC activity, e.g., downregulated *Mpc1/2* expression, lower MPC1/2 protein abundances or reduced carrier activity, is needed to optimize MPC-targeted therapy.

### Pyruvate dehydrogenase complex

PDC catalyzes the irreversible oxidative decarboxylation of pyruvate to form acetyl-CoA, $CO_2$ and NADH. This conversion is normally the major fate of the transported pyruvate into the mitochondrial matrix in order to support ATP production by OXPHOS. Therefore, PDC holds a pivotal role in the oxidation of glucose by connecting the glycolytic pathway in the cytosol with the TCA cycle activity and OXPHOS in mitochondria. With this, PDC functions as a regulator in the metabolism of pyruvate to maintain glucose homeostasis (Patel and Korotchkina, 2006). In addition, PDC activity is also crucial for providing acetyl-CoA for FFA synthesis (Sugden and Holness, 2006).

PDC is a large multi-component enzymatic complex (10 MDa) comprising of three catalytic, one binding and two regulatory components: pyruvate dehydrogenase (PDH or E1), dihydrolipoamide acetyltransferase (E2), dihydrolipoamide dehydrogenase (E3), E3 binding protein (E3P, protein X), pyruvate dehydrogenase kinase (PDK) and pyruvate dehydrogenase phosphatase (PDP), respectively. The core structure consists of E2 and E3BP, which in turn enlist E1 and E3. The three catalytic components, E1–E3, operate the conversion of pyruvate in a sequential manner (Smolle et al, 2006). First, the rate-limiting step is mediated by E1 which is a heterotetrameric complex comprising two copies of each of the E1α and E1β subunits, with each PDC containing 20–30 E1 complexes. The E1 is a thiamine diphosphate-dependent enzyme responsible for the oxidative decarboxylation of pyruvate which is coupled to the reductive acetylation of a lipoamide cofactor. Subsequently, this acetyl group is transferred from thiamine pyrophosphate to a lipoate moiety covalently attached to E2. In turn, E2 catalyzes the transfer of this acetyl group from the lipoate moiety to CoA resulting in the formation of acetyl-CoA and dihydrolipoate. The E2 comprises four domains namely an inner domain, a subunit binding domain, and two lipoyl domains (L1 and L2) and each PDC consists of 40–42 E2 subunits (Vijayakrishnan et al, 2010). Finally, E3 is responsible for the regeneration of the lipoate group via the oxidation of dihydrolipoate coupled with the reduction of FAD to $FADH_2$. Subsequently, $FADH_2$ is re-oxidized by $NAD^+$ resulting in the formation of NADH and the regeneration of FAD. The core structure of PDC is associated with 6–12 E3 homodimeric subunits via the interaction at the subunit binding domain of two E3BP proteins. The latter is a crucial, structural protein of the core structure of PDC with no enzymatic activity. Structurally, E3BP bears a resemblance to E2 as it also includes an inner domain, a subunit binding domain and one lipoyl domain (L3). Each PDC contains 18–20 E3BP proteins (Fig. 5) (Patel and Korotchkina, 2006; Gray et al, 2014).

Considering the pivotal role of PDC in maintaining cellular energy homeostasis, the flux through PDC is tightly regulated at different levels depending on the energy status of the cell. Fasting, for example, results in reduced expression of transcripts for PDC proteins (long-term regulation) (Zhang et al, 2011). However,

short-term regulation via the reversible phosphorylation/dephosphorylation of three serine residues on the E1α subunit, namely Ser-264 (site 1), Ser-271 (site 2) and Ser-203 (site 3) in humans or Ser-293 (site 1), Ser-300 (site 2) and Ser-232 (site 3) in mice, is the primarily and most critical mechanism in controlling PDC activity (Fig. 5) (Patel and Korotchkina, 2006). Phosphorylation of one site/serine residue, with site 1 as the most common target, is already adequate to disable enzymatic activity as p-serine residues lose their capacity to bind and bring the lipoyl domains to the active site of E1 (Kato et al, 2008). This serine phosphorylation is mediated by four isoforms of PDKs (PDK1-4) and its dephosphorylation is conducted by two isoforms of PDPs (PDP1 and PDP2). In mammals, all PDKs and PDPs are located in the mitochondrial matrix and show a tissue-specific expression pattern for tissue-specific regulation of PDH. For PDKs, PDK1 is mainly present in the heart, with reduced levels in the skeletal muscle, liver and pancreatic islets. PDK2 is ubiquitously present in many tissues. PDK3 is mostly distributed in the testis, with lower levels in the lung, brain and kidney. High PDK4-expressing tissues are the skeletal muscle, liver, kidney and pancreatic islets (Wang et al, 2021). Furthermore, all four isoforms exhibit distinct preferences and phosphorylation rates at the three sites of PDH. PDK1 can phosphorylate all three sites with a preference order of site 1 > site 3 > site 2. In contrast, the other PDKs (PDK2-4) only phosphorylate site 1 and 2 of PDH. PDK2 shows a preference for phosphorylating site 1 over site 2. Therefore, PDK2 has less phosphorylation capacity for site 2 compared to PDK3 and PDK4, with PDK4 displaying the highest activity towards site 2. PDK3 phosphorylates site 1 faster than site 2. Overall, the phosphorylation potential of PDH by all PDKs ranks as follows: PDK1 > PDK3&4 > PDK2 (Korotchkina and Patel, 2001; Kolobova et al, 2001). On the other hand, PDP1 is highly detected in the heart, brain, skeletal muscle and testis while PDP2 is mostly present in the liver, kidney, brain, heart and adipose tissue (Huang et al, 2003). The dephosphorylation rate is the highest for site 2 and the lowest for site 1, but both PDP1 and PDP2 can act on all three phosphorylation sites with PDP1 showing the highest activity compared to PDP2 (Patel and Korotchkina, 2006). As stated before, the energetic status of the cell determines the activity status of PDH via the present levels of PDKs and PDPs. During normal, fed conditions, PDPs are increased and activate PDH to induce pyruvate oxidation and ATP production to meet the energy demands of the cell. However, during starvation, PDK transcription is increased while PDP expression is decreased resulting in diminished carbohydrate oxidation and increased GNEO to maintain blood glucose levels (Gray et al, 2014; Wang et al, 2021). Besides this, PDK and PDC activity are also regulated by the levels of metabolites that serve as a substrate for PDC. High amounts of pyruvate, ADP, $NAD^+$ and CoA inhibit PDK while increased levels of ATP, NADH, and acetyl-CoA activate PDK, but inhibit PDP (Prochownik and Wang, 2021).

As sepsis is characterized by a failing starvation response, a lot of research has been conducted toward PDC and PDK activity in sepsis and toward its role in lethality (Vandewalle and Libert, 2022). One of the first studies performed by Vary et al demonstrates that skeletal muscle tissue of septic (small or large septic abscess) rats is characterized by a threefold reduced proportion of the active PDC relative to control rats. This is correlated with a change in skeletal muscle acetyl-CoA/CoA ratio, increased PDK activity and hyperlactatemia (Vary and Siegel,

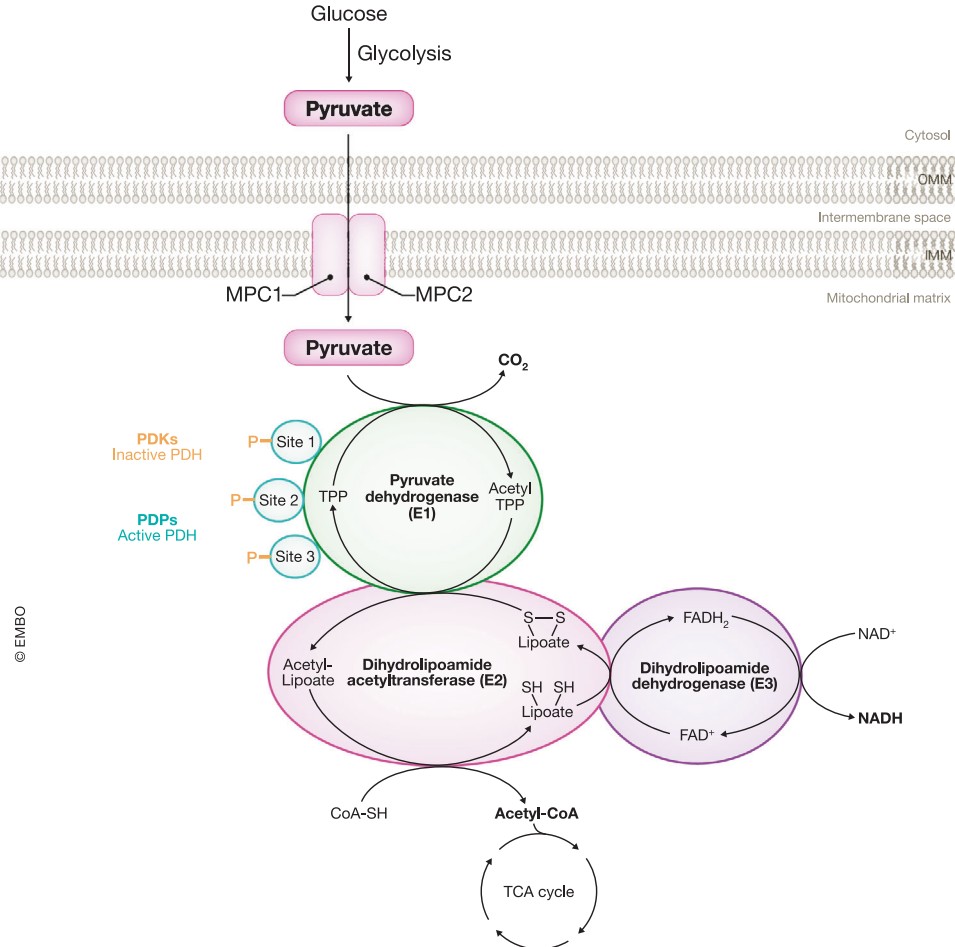

**Figure 5. The Overall reaction mechanism and regulation of the pyruvate dehydrogenase complex.**

Briefly, glucose undergoes conversion into pyruvate through glycolysis and the resultant pyruvate is transported across the inner mitochondrial membrane (IMM) by the mitochondrial pyruvate carrier (MPC). The three catalytic components of the pyruvate dehydrogenase complex, i.e., pyruvate dehydrogenase (E1, depicted in green), dihydrolipoamide acetyltransferase (E2, depicted in pink), and dihydrolipoamide dehydrogenase (E3, depicted in purple), sequentially facilitate the oxidative decarboxylation of pyruvate, leading to the production of acetyl-CoA, $CO_2$ and NADH. The predominantly regulatory mechanism governing the pyruvate dehydrogenase complex is the reversible phosphorylation/dephosphorylation of three serine residues (Site 1–3) on E1 by pyruvate dehydrogenase kinases (PDKs) and pyruvate dehydrogenase phosphatases (PDPs), respectively. IMM inner mitochondrial membrane, OMM outer mitochondrial membrane, TPP thiamine pyrophosphate, $FADH_2$ flavin adenine dinucleotide, NAD(H) nicotinamide adenine dinucleotide (+ hydrogen), CoA coenzyme A, TCA tricarboxylic acid cycle.

1989; Vary et al, 1996; Vary and Hazen, 1999). Moreover, peripheral blood mononuclear cells of septic patients exhibit diminished PDC levels and activity compared to healthy controls which was associated with reduced survival chance (Nuzzo et al, 2015). Interestingly, targeting PDK with the prototypic drug, dichloroacetate (DCA, a structural analog of pyruvate), lowers phosphorylation on the E1α subunit which ultimately improves PDH activity and lowers blood lactate levels. Indeed, DCA treatment improves oxygen consumption, pyruvate oxidation and reduces plasma lactate concentration in septic patients (Stacpoole et al, 1992; Gore and Demaria, 1996). However, the clinical trial of Stacpoole et al shows that DCA treatment fails to improve the survival rate (Stacpoole et al, 1992). Despite this, DCA protection has been widely observed in many studies using sepsis models. This DCA-mediated protection mechanism occurs at multiple levels. Firstly, DCA treatment alleviates the metabolic and energy dysfunctions observed during sepsis. Hepatocellular metabolic

dysfunction enzyme markers such as ALT and aspartate amino-transferase (AST) are markedly decreased in septic mice (McCall et al, 2018). Furthermore, sepsis-induced anorexia along with reduced food intake are reverted after DCA administration which is accompanied with a restoration in systemic fuel availability. The latter is associated with septic systemic metabolic alterations including a partially reduced sepsis-induced hypoglycemia and FFA levels (McCall et al, 2018; Oh et al, 2022). In addition, DCA stimulates the mitochondrial oxidative metabolism as isolated septic hepatocytes and splenocytes show increased mitochondrial respiration and energy index. This is accompanied with a reversal in hepatic transcriptional changes of genes involved in mitochondrial dysfunction and with a reversal to normal TCA metabolite levels compared to control levels (Mainali et al, 2021; Oh et al, 2022). As mentioned before, sepsis is associated with increased lipolysis and hepatic steatosis and DCA treatment could restore this based on lipidomics analysis in septic mice (Mainali et al, 2021).

Notably, apart from the liver and skeletal muscle, the heart tissue is also characterized by PDC inactivation which is associated with SIC. A proteomics study of septic mice hearts by Shimada et al reveals a changed mitochondrial metabolic protein profile involving pyruvate metabolism, lactate production, fatty acid metabolism, electron transport and mitochondrial membrane integrity. Interestingly, alterations in pyruvate metabolism proteins is accompanied with increased PDK4 levels resulting in elevated phosphorylation and inactivation of PDH which is associated with a reduced pyruvate-driven OXPHOS by cardiac mitochondria of septic mice (Shimada et al, 2022). In addition, Chen et al demonstrates that PDK4 upregulation in LPS-treated cardiomyocytes leads to mitochondrial damage by promoting lactate production due to PDH inhibition, eventually contributing to SIC (Chen et al, 2024). Alongside this, increased serum PDK4 levels are observed in pediatric patients with SIC and this is coupled with increased disease severity and mortality. PDK4 cardiac-specific knockdown or inhibition in septic mice mitigates myocardial injury and mitochondrial dysfunction and enhances myocardial contractile function (Chen et al, 2023).

Finally, DCA treatment also improves the immune response and competence observed during sepsis. For instance, septic mice display a significantly elevated blood lymphocyte count and a repolarization of splenic effector and repressor immune cell response upon DCA administration. This is accompanied with increased levels of pro-inflammatory cytokines (e.g., IL-12), decreased IL-10 levels and enhanced peritoneal bacterial clearance (McCall et al, 2018). Furthermore, Meyers et al provide evidence that PDK maintains NLRP3 inflammasome activation in murine and human macrophages as PDK inhibition reduces NLRP3 inflammasome-mediated inflammation and cell death. This inhibition is concurrent with a shift in metabolic fueling and energy homeostasis, increasing autophagic flux, inducing mitochondrial fusion, fitness and structure, and reducing ROS production in NLRP3 inflammasome-activated macrophages (Meyers et al, 2023).

In addition, thiamine (vitamin B1) is an essential cofactor for PDH activity. Notably, thiamine deficiency is highly prevalent, affecting 71.4% patients with septic shock, where low thiamine levels are associated with organ dysfunction and lactic acidosis (Donnino et al, 2010; Costa et al, 2014). The latter is likely due to pyruvate accumulation resulting from PDH inactivity. Interestingly, numerous clinical trials demonstrate the beneficial effects of thiamine administration in alleviating hyperlactatemia and reducing mortality in sepsis patients (Woolum et al, 2018; Harun et al, 2019). Hence, it could be considered to provide thiamine supplementation to septic shock patients in the ICU (Singer et al, 2009).

In summary, septic subjects exhibit inactive PDH due to heightened PDK activity, correlating with elevated mortality rates. Administration of DCA holds significant therapeutic promise for sepsis as it mitigates various metabolic dysfunctions, including hyperlactatemia, hypoglycemia, mitochondrial oxidative metabolism, elevated lipolysis and hepatic steatosis. In addition, DCA improves myocardial injury in SIC and results in an enhanced immune response, thereby facilitating bacterial clearance and ultimately promoting homeostasis and sepsis survival. However, clinical trials studying the efficacy of DCA in septic patients remain limited. DCA treatment could effectively reduce hyperlactatemia, but fails to improve the survival rates (Stacpoole et al, 1992). Nevertheless, further exploration of the protective role of DCA in sepsis patients is strongly recommended. In contrast, thiamine has been heavily studied in several clinical trials and shows promising

therapeutic effects on blood lactate levels and survival rates. However, further investigation is still needed to elucidate the mechanism of thiamine protection and to correlate this with increased PDH activity in septic individuals.

### Pyruvate carboxylase

PC catalyzes the irreversible, MgATP-dependent carboxylation of pyruvate and $HCO_3^-$ into oxaloacetate. This oxaloacetate can anaplerotically be used to replenish the TCA cycle or it can be withdrawn outside the mitochondria to fuel GNEO. In addition, the formed oxaloacetate via PC activity is also important for lipogenesis, insulin secretion and neurotransmitter production. Hence, PC functions as a crucial enzyme in intermediary metabolism (Jitrapakdee et al, 2008; Adina-Zada et al, 2012).

PC is part of the family of biotin (Vitamin B7)-dependent carboxylases and is encoded by the *Pc* gene in humans and *Pcx* gene in mice. It is ubiquitous expressed with increased levels in gluconeogenic tissues such as the liver and kidneys, adipose tissue, pancreatic islets, and the heart. PC is a multisubunit enzyme comprised of four identical subunits ($\pm$ 120–130 kDa each) organized as two dimers that bind in an anti-parallel manner in order to create an enzymatically active homotetramer (Jitrapakdee et al, 2008). This is crucial as the monomeric PC form has no activity. Each subunit contains three functional domains, starting from the N-terminus: a biotin carboxylase (BC) domain, a carboxyltransferase (CT) domain, and a biotin carboxyl carrier protein (BCCP) domain containing a covalently bound biotin cofactor at a specific lysine residue (Adina-Zada et al, 2012). The pyruvate carboxylation reaction occurs in two different steps. First, the BC domain results in the formation of a carboxyphosphate intermediate via a MgATP-dependent phosphorylation of $HCO_3^-$ which then transfers the activated carboxyl group from the carboxyphosphate intermediate to the tethered biotin cofactor creating carboxybiotin. Next, the carboxybiotin is translocated to the CT domain where the carboxyl group is transferred to pyruvate, creating oxaloacetate and regenerating the biotin cofactor (Jitrapakdee et al, 2008; Adina-Zada et al, 2012).

The activity of PC is strictly controlled as it plays a pivotal role in the metabolic intersection of carbohydrate and lipid metabolism. The most important regulation is via the allosteric activator, acetyl-CoA, which binds to PC creating a fully stabilized, catalytically active tetrameric form (Jeoung et al, 2014; Chai et al, 2022). High levels of acetyl-CoA depend on the energetic status of the cell and are mainly produced via increased PDH activity and FFA oxidation. This means that on the one hand, cellular energy needs are satisfied and GNEO should be induced or on the other hand, that there is an energy deficit and there is a need for anaplerotic replenishment of the TCA cycle via PC activity, respectively. The coordinated regulation of PC and PDH determines the overall direction of carbon flux (Gray et al, 2014). Besides acetyl-CoA, ATP is also an allosteric activator while glutamate and aspartate are allosteric inhibitors (Gray et al, 2014; Jeoung et al, 2014). In addition to allosteric regulation, PC is also subjected to transcriptional regulation. Upstream stimulatory factors, peroxisome proliferator-activated receptor-γ, hepatocyte nuclear factor 3β, or forkhead/winged helix transcription factor box 2 are known regulators of PC expression (Boonsaen et al, 2007).

Since PC is a key enzyme in pyruvate-driven GNEO, and sepsis is known for causing disturbances in GNEO, it is interesting to consider if sepsis influences PC activity. Unfortunately, there is

limited research into the role of PC activity in sepsis. Nevertheless, studies examining renal and hepatic GNEO flux in septic rats have revealed a significant reduction in the activity of key metabolic enzymes involved in GNEO, including PC (Ardawi et al, 1990; Jones and Titheradge, 1993). Jones et al even show a 50% decrease in hepatic PC activity during sepsis (Jones and Titheradge, 1993). Furthermore, a study into the effects of LPS on blood glucose and hepatic GNEO in dairy goats reveals a marked decrease in blood glucose levels and a reduced expression of PC (Wang et al, 2015). Interestingly, pancreatic islet β-cells also exhibit reduced PC activity when treated with pro-inflammatory cytokines (TNFα, IL-1β, and IFN-γ). Increasing PC activity, and thus PC-mediated TCA anaplerosis, protects β-cells against inflammation toxicity by increasing aspartate availability through oxaloacetate conversion via aspartate aminotransferase. Elevated aspartate creates an argininosuccinate shunt, activating the urea cycle and redirecting arginine metabolism towards ureagenesis. This limits arginine use for inflammation-induced NO synthesis, reducing oxidative damage and preventing cell death of pancreatic islet β-cells (Fu et al, 2020). PC activity can also facilitate an extra protective mechanism against oxidative stress by controlling the NADPH/NADP$^+$ ratio through sustaining malic enzyme activity. Liver-specific PC knockout (LPCKO) mice show diminished malate production and accumulation of pyruvate, resulting in altered malic enzyme activity leading to NADPH/NADP$^+$ depletion, diminished antioxidant defenses via a reduced glutathione antioxidant mechanism, and increased liver inflammation susceptibility (Cappel et al, 2019). In addition to preserving the glutathione antioxidant activity, PC is crucial for de novo glutathione synthesis, thereby limiting inflammation-induced ROS accumulation in pancreatic islet β-cells during inflammatory stress (Fu et al, 2021). Conversely, a recent study by Liang et al demonstrated increased PC activity in in vitro (LPS-treated colorectal cells and LPS-treated monocytes) and in vivo (Dextran sulfate sodium (DSS)-induced) models of colitis. This was accompanied with elevated NF-κB signaling, increased production of inflammatory cytokines and ROS, decreased pyruvate content, and accumulation of acetyl-CoA, oxaloacetate and lactate. Moreover, treatment with anemoside B4, a specific PC inhibitor, could ameliorate the inflammatory response, oxidative stress, and could regulate the levels of pyruvate and its downstream metabolites. These findings suggest that PC-mediated changes in pyruvate and acetyl-CoA levels are essential in mediating the inflammatory response and ROS production in LPS-treated colorectal cells. Hence, targeting PC with anemoside B4 might be a potential anti-inflammatory approach (Liang et al, 2024). Notably, PC activity is also crucial in viral innate immune responses by targeting the RIG-I-MAVS-TRAF6-signaling pathway. This promotes NF-κB activation and increases production of interferons and pro-inflammatory cytokines in viral-infected cells (Cao et al, 2016). Moreover, a metabolomics analysis of SARS-CoV-2 infected kidney epithelial cells and lung air-liquid interface cultures, demonstrates that SARS-CoV-2 modifies the TCA cycle metabolism. This is associated with redirecting glucose-derived carbon entry into the TCA cycle through elevated PC activity and with a significant reduction in glutaminolysis (Mullen et al, 2021).

Considering the crucial role of PC activity in metabolism, it comes as no surprise that PC activity plays an essential role during altered hepatic metabolic states, for example, in starvation and T2D.

Interestingly, LPCKO mice show a loss in hepatic GNEO capacity, elevated blood lactate levels and increased ketogenesis. However, LPCKO mice are able to survive a 24 h fast whereby they maintain physiologically normal glucose levels indicating a compensation by other GNEO tissues and that the liver is depending on alternative fuel sources (Cappel et al, 2019; Selen et al, 2022). In addition, inhibiting hepatic PC expression by a specific antisense oligonucleotide has been shown to reduce plasma glucose levels, decreased hepatic steatosis, and enhance hepatic insulin sensitivity in diabetic rats induced by a high-fat diet (HFD) (Kumashiro et al, 2013). Moreover, LPCKO mice are also protected against HFD-induced hyperglycemia and insulin resistance, but it does not prevent hepatic steatosis. (Cappel et al, 2019; Selen et al, 2022). Furthermore, it should be noted that LPCKO mice manifest a clear metabolic dysfunction characterized by aggravated hypoglycemia and hyperlactatemia when fed with a carbohydrate-limited ketogenic diet. This shows the importance of PC activity in controlling glucose levels during carbohydrate-restricted conditions (Selen et al, 2022).

Obviously, further research is warranted as there is a clear discrepancy in PC activity during inflammatory diseases and sepsis. However, it is speculated that PC activity is impaired in septic subjects based on the following assumptions: (1) Previously mentioned studies show reduced hepatic GNEO flux associated with decreased PC activity in septic rats. (2) Sepsis is characterized by a failing starvation response resulting in impaired PDC activity and FFA oxidation, likely leading to decreased acetyl-CoA levels thereby inactivating PC. (3) Hypoglycemia is a common feature in septic mice and as PC is a crucial enzyme in pyruvate-driven GNEO, impaired PC activity could be at the basis of this phenomenon. (4) Hyperlactatemia observed in septic individuals could be explained by reduced PC activity, as demonstrated in LPCKO mice showing increased blood lactate levels. Hence, exploring the effects of LPCKO or PC inhibition by AB4 on these sepsis-associated metabolic alterations could be of interest. On the other hand, it should also be considered to study the anti-inflammatory actions of AB4 on sepsis survival.

## Malic enzyme

MEs are oxidoreductases that catalyze the oxidative decarboxylation of L-malate to pyruvate and $CO_2$ with the simultaneous reduction of NAD(P)$^+$ to NAD(P)H. It functions as a metabolic node connecting glycolysis with the TCA cycle, and it is also important in glutamine metabolism and GNEO. Moreover, NADPH production is essential for lipid biosynthesis and redox homeostasis (Prochownik and Wang, 2021). In mammals, three different ME isoforms have been identified: (1) cytosolic, NADP$^+$-dependent ME (ME1), mitochondrial NADP$^+$-dependent ME (ME2), and mitochondrial NADP$^+$-dependent ME (ME3). This distinction is made based on subcellular localization and cofactor use. Each isoform is a homotetrameric protein with a double dimer structure, wherein the dimer interface promotes a more robust interaction compared to the tetramer interface. Each monomer is composed out of four structural domains (A, B, C, and D), and the active site of the enzyme lays on the interface between domains B and C (Chang and Tong, 2003). Furthermore, the activity of MEs is mainly controlled by transcriptional regulation, posttranslational modifications such as phosphorylation and acetylation, and allosteric regulation with fumarate as an activator and ATP as an inhibitor (Chang and Tong, 2003). However, studies on the role of

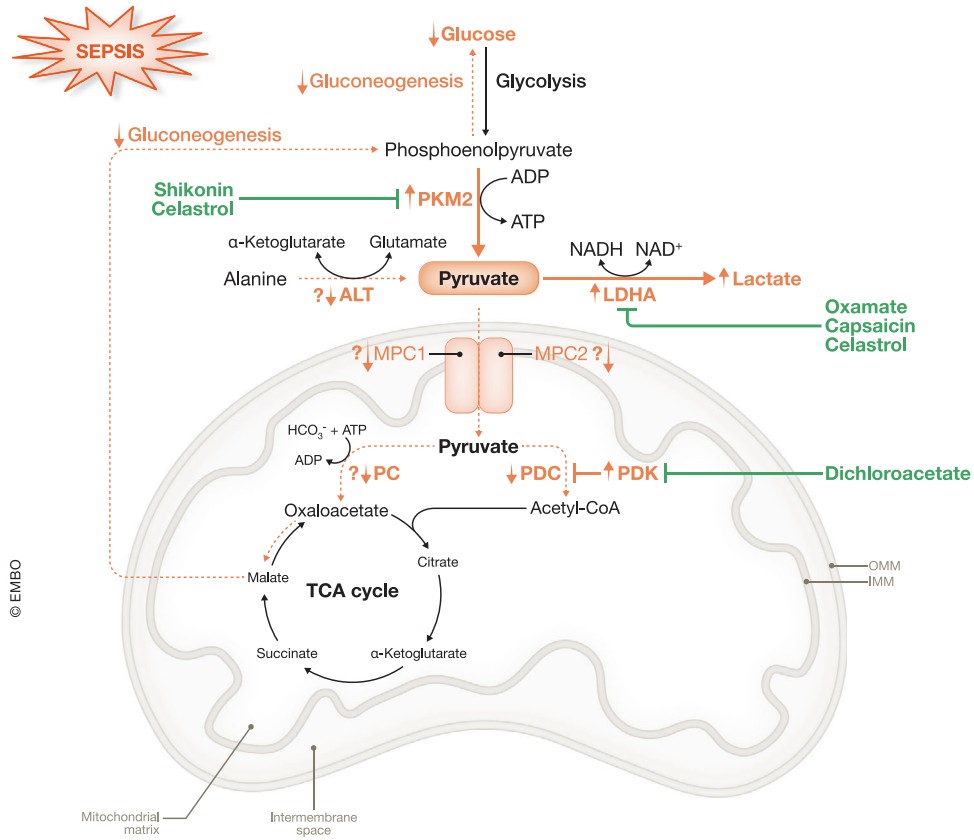

**Figure 6. Therapeutic strategies to prevent altered pyruvate metabolism in sepsis.**

Sepsis is characterized by a dysregulated pyruvate metabolism (indicated by the orange arrows). First, an elevated glycolytic metabolic flux is induced by increased pyruvate kinase M2 (PKM2) and lactate dehydrogenase A (LDHA) activity leading to enhanced lactate levels. Second, potential disturbances in alanine aminotransferase (ALT), mitochondrial pyruvate carrier (MPC), and pyruvate carboxylase (PC) activity could be involved in the disrupted gluconeogenesis pathway, eventually responsible for hypoglycemia. Third, problems in mitochondrial pyruvate-driven tricarboxylic acid (TCA) cycle rate and oxidative phosphorylation capacity are the result of diminished pyruvate dehydrogenase complex (PDC) activity due to the inhibitory effects of pyruvate dehydrogenase kinases (PDKs) or by potential disturbances in MPC activity. Finally, an overview of the therapeutic drugs targeting enzymes involved in pyruvate metabolism and that can improve sepsis survival, are indicated in green. ADP adenosine diphosphate, ATP adenosine triphosphate, IMM inner mitochondrial membrane, OMM outer mitochondrial membrane, NAD(H) nicotinamide adenine dinucleotide (+ hydrogen).

ME activity in sepsis are currently limited. Despite this, it should be insightful to investigate potential disturbances in ME activity in septic individuals and its potential role in sepsis metabolic alterations, such as impaired GNEO resulting in hypoglycemia.

## General conclusions

It is evident that disturbances in enzyme activity involved in pyruvate metabolism play a key role in metabolic reprogramming and survival during sepsis progression. First of all, the Warburg phenotype in activated immune cells is a generally accepted phenomenon in sepsis, mediated via increased activity of PKM2 and LDHA. With this, improved survival outcomes have been shown with specific inhibitors, i.e., celastrol and shikonin for PKM2 and celastrol, sodium oxamate and capsaicin for LDHA, alleviating increased glycolytic flux and decreasing lactate production and its toxic effects. However, the PKM2-mediated cardioprotective effects during SIC and the importance of LDHA-mediated Warburg

phenotype in septic PMNs functioning should be considered in further therapeutic developments targeting PKM2 and LDHA (Fig. 6 and Table 1). Second, septic individuals exhibit hepatic GNEO failure and hypoglycemia which is linked to a higher mortality. In line with this, it is postulated that alanine-driven GNEO is disrupted due to decreased hepatic ALT activity, and more importantly that pyruvate-driven GNEO is disturbed due to malfunctioning of hepatic mitochondrial MPC and/or PC resulting in enhanced lactate production (Fig. 6). However, further research is necessary to completely explore this potential reduction in activity of these metabolic enzymes and mitochondrial transporter in sepsis. Finally, mitochondrial pyruvate-driven oxidative metabolism is compromised in sepsis resulting in increased cytosolic lactate production. This is primarily caused by hampered PDC activity due to PDK phosphorylation and/or potential disturbances in MPC function. Notably, treatment with a PDK-specific inhibitor, DCA, ameliorates sepsis survival at multiple levels: (1) it augments metabolic dysfunctions, e.g., alleviates mitochondrial respiration and lactate production, (2) it impairs hepatic steatosis, (3) it

**Table 1. Overview of the effects of modified enzyme activities in pyruvate metabolism on in vivo sepsis outcomes.**

| Cell type | Effects | References |
|---|---|---|
| **Upregulation of PKM2** | | |
| Macrophages | Detrimental:<br>- Increased aerobic glycolysis and lactate production due to HIF1α activation. The formed lactate results in inflammasome activation and pro-inflammatory mediator release.<br>- IL-1b production. | Yang et al, 2014; Palsson-Mcdermott et al, 2015; Xie et al, 2016 |
| Cardiomyocytes | Beneficial:<br>- Interaction with SERCA2a for maintaining cardiac calcium homeostasis and cardiac contraction, thereby reducing myocardial injury.<br>- Attenuating mitochondrial damage by regulating mitochondrial biogenesis in a prohibitin 2 dependent way. | Ni et al, 2022; Ren et al, 2024; Du et al, 2024 |
| **Upregulation of LDH** | | |
| Systemic | Detrimental:<br>- Hyperlactatemia | Levy et al, 2005; Garcia-Alvarez et al, 2014; Tan et al, 2021; Yang et al, 2022; Zhang et al, 2022; Pan et al, 2022 |
| Macrophages | Detrimental:<br>- Lactate – GPR81 signaling results in increased HMGB1 release, thereby contributing to endothelial barrier dysfunction.<br>- Increasing glycolytic flux and the Warburg effect-mediated inflammatory storm. | Luo et al, 2022; Yang et al, 2022; Zhang et al, 2022 |
| **Downregulation of LDH** | | |
| Neutrophils | Detrimental:<br>- Reduced chemotaxis and phagocytosis capacities due to a diminished Warburg phenotype. | Pan et al, 2022 |
| **Downregulation of PDH by activated PDKs** | | |
| Systemic | Detrimental:<br>- Sepsis-induced anorexia and reduced food intake<br>- Hyperlactatemia<br>- Hypoglycemia<br>- Increased lipolysis & elevated FFA levels<br>- Imbalance innate and adaptive immune response<br>- Reduced clearance of infecting organisms | McCall et al, 2018; Mainali et al, 2021; Oh et al, 2022 |
| Hepatocytes | Detrimental:<br>- Reduced mitochondrial oxidative metabolism and energy index<br>- Hepatic steatosis | McCall et al, 2018; Mainali et al, 2021; Oh et al, 2022 |
| Cardiomyocytes | Detrimental:<br>- Reduced pyruvate-driven OXPHOS<br>- Mitochondrial damage by lactate production<br>- Increased myocardial injury & lower myocardial contractile function | Shimada et al, 2022; Chen et al, 2023; Chen et al, 2024 |
| Macrophages | Detrimental effects of activated PDKs:<br>- NLRP3 inflammasome activation<br>- Metabolic rewiring<br>- Reduced autophagic flux<br>- Reduced mitochondrial fusion<br>- Influencing mitochondria structure<br>- Increased ROS production | McCall et al, 2018; Meyers et al, 2023 |

reduces SIC, and (4) it enhances the immune response (Fig. 6 and Table 1). Overall, research into the dysregulation of both cytosolic and mitochondrial pyruvate metabolic enzymes has uncovered several beneficial therapeutic strategies for enhancing sepsis survival. Nonetheless, further exploration of these pyruvate metabolism-related metabolic shifts remains crucial and potentially holds much promise in terms of therapeutic successes and patient outcomes.

## Pending issues

i. Research towards the role of alanine aminotransferase, mitochondrial pyruvate carriers and pyruvate carboxylase in metabolic alterations and inflammatory responses in septic animals and human patients.

ii. Clinical studies to validate mouse sepsis data on altered enzyme activities in pyruvate metabolism, beyond mRNA and protein quantities.

iii. Clinical trials and stratification based on pyruvate metabolism to determine therapy is needed.

## Peer review information

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

## Acknowledgements

Research in the author's laboratory was supported by the following FWO Flanders grants: "Investigating the role of lactate-mediated lethal shock in sepsis, Louise Nuyttens, 11M3122N"; "Investigating the protective role of the glucocorticoid receptor against lactate-mediated lethal shock in sepsis, Jolien Vandewalle, 1220924N"; "Investigation of the mechanism of the loss of function of HNF4a-PPARa axis in polymicrobial sepsis, Claude Libert, 3G014921"; "STOP_SEPSIS: investigation into a new concept of sepsis, the greatest unmet medical need of our times, Claude Libert, S003122N" and UGent grant "Stop Sepsis, Claude Libert, 01M00121".

## Author contributions

**Louise Nuyttens**: Writing—original draft; Writing—review and editing. **Jolien Vandewalle**: Writing—review and editing. **Claude Libert**: Writing—review and editing.

## Disclosure and competing interests statement

Prof. Claude Libert is a member of the *EMBO Molecular Medicine* Editorial Board. This has no bearing on the editorial consideration of this article for publication. The remaining authors declare no competing interests.

