## [Peer Review File · EMBO Molecular Medicine]

Sepsis-induced changes in pyruvate metabolism: insights and potential therapeutic approaches.

Louise Nuyttens, Jolien Vandewalle, and Claude Libert

Corresponding author(s): Claude Libert (claude.libert@irc.vib-ugent.be)

Review Timeline:

Submission Date:	10th Jun 24
Editorial Decision:	26th Jun 24
Revision Received:	30th Aug 24
Editorial Decision:	18th Sep 24
Revision Received:	24th Sep 24
Accepted:	26th Sep 24

Editor: Lise Roth

Transaction Report:

26th Jun 2024

Dear Claude,

Thank you for the submission of your review to EMBO Molecular Medicine. We have now received feedback from the experts who agreed to evaluate your manuscript.

As you will see from the reports below, they overall found the review well written, interesting, and timely. They nevertheless raise a few concerns and make several suggestions to improve the interest and impact of your work.

We would therefore welcome a revised version of your manuscript that would address these points. Please attach a covering letter giving details of the way in which you have handled each of the points raised by the referees.

- 1/ A .doc formatted version of the manuscript text (including Figure legends and tables)
- 2/ Separate figure files
- 3/ A letter INCLUDING the reviewer's reports and your detailed responses to their comments.
- 4/ A glossary: EMBO Molecular Medicine articles are accompanied by a glossary explaining some of the terms used for laymen.
- 5/ Pending issues: At the end of each article, there is a box highlighting issues that still need further studies and where research efforts should converge (called the Pending issues box).
- 6/A 'disclosure statement and competing interests' statement (<https://www.embopress.org/competing-interests>).

For the figures:

We work with one of our expert scientific illustrators, who will assist with getting the figure to a publication ready state. What we need from you is a draft that accurately illustrates the key scientific concepts that you wish to show.

Please also note the following points:

- If there are certain aspects of your figure draft that are based upon assumptions or where the scientific data remains ambiguous, please add a comment so that we can work with you on an accurate depiction. Please ensure the directionality and nature of interactions is presented accurately.
- If the figure or single panels of the figure have been adapted from a published figure, please add this information to the figure legend (e.g., 'Adapted from...' or 'Based on...').
- Please only re-use figures or parts of a figure if this is essential for understanding the concept communicated. Often a reference to a previous paper will suffice. If the figure contains re-used images or elements of images, please make sure that you have the permission/license to publish it (this also applies to your own previous work, if the journal you published in retains copyright. Certain 'creative commons' open access licenses, such as CC-BY 4.0, allow re-use without additional formal permissions). All re-used material must be explicitly cited.
- If you use an image data base for scientific iconography (e.g., BioRender), please let us know if you have a license that allows for publication in an academic journal. Often authors use misleading iconography for expedience. Please ensure the information shown is scientifically accurate.
- For figures created using a software for editing vector objects like Inkscape, CorelDraw etc., please send the file as a PDF (or SVG, or EPS), PowerPoint or Keynote in which the labels and objects are still editable. For figures created using Adobe Illustrator, please send the Illustrator (.ai) file.

Looking forward to receiving your revised manuscript,

With kind regards,

Lise

***** Reviewer's comments *****

Referee #1 (Remarks for Author):

This paper overviews the cytosolic and mitochondrial enzymes involved in pyruvate metabolism and how their activities are disrupted in sepsis. Therapeutic strategies targeting these pyruvate-related enzymes in sepsis are also discussed.

General Comments:

This review nicely overviews the pathways involved in pyruvate metabolism. The concept is that pyruvate is the only source of lactate in mammals, and that accumulation of lactate is a biomarker of sepsis severity. The paper provides a comprehensive description of the different pathways involved in pyruvate metabolism, what happens to these pathways in sepsis, and what is known about targeting these pyruvate metabolic pathways to treat sepsis. However, I believe the authors need to better delineate how important modifying the pyruvate metabolic pathways are in modifying lactate levels, versus altering other metabolic and inflammatory processes.

In sepsis, it is not only the accumulation of lactate that is harmful, but also the development of acidosis that results from glycolysis uncoupled from pyruvate oxidation or carboxylation, versus pyruvate destined for lactate production. This concept is largely ignored in this review, and I believe needs to be addressed.

Specific Comments:

- 1) While a detailed discussion is provided on lactate production from LDH in sepsis, there is almost no discussion of the fate of the lactate. Recent studies have suggested that inhibition of the monocarboxylic acid transporter (MCT) can favourably affect pyruvate fate in muscle. I believe some discussion of the potential of MCT inhibition in sepsis is warranted.
- 2) Pyruvate dehydrogenase I (PDH) activity is markedly affected by fatty acids, which can increase dramatically in sepsis. The potential role of increased fatty acid oxidation in inhibiting PDH in sepsis should be discussed.
- 3) I am presuming that "GNEO" stands for gluconeogenesis. This could be more clearly indicated in the manuscript.
- 4) It is stated that PK is the rate-limiting enzyme of glycolysis. This is not universally accepted, and many studies propose phosphofructokinase is the rate-limiting enzyme for glycolysis.
- 5) I believe the role of PEPCK in gluconeogenesis in sepsis is worth discussing.
- 6) It is stated that "Unfortunately, research towards PC activity in sepsis is currently constrained." What does this mean?
- 7) It is not clear how important gluconeogenesis is in sepsis, and whether inhibiting or stimulating gluconeogenesis would be desirable.

Referee #2 (Remarks for Author):

Nuytens et al. review sepsis-induced changes in pyruvate metabolism and their therapeutic implications. This is an in-depth and authoritative manuscript. Here are a few major points that, in this reviewer's opinion, are critical to improving the breath, appeal, and relevance of this work:

1. The review is centered on Sepsis, which is a condition that is so relevant because it is a leading cause of human mortality and disability, accounting for ~20% of all global human mortality. However, the authors chose mouse models that have severe limitations to satisfactorily model the condition as the focus of the review, only citing human data that supports mouse observations. It would be more significant to do the exact opposite: review human data and cite mouse data when that has mechanistic value to understand the observations. As done here, it runs into the problem that often very promising and exciting observations do not translate into progress in therapeutic interventions. A good example in the current work is DCA, which has shown disappointing results in clinical trials already several decades ago. In this context, it would be useful that the authors describe and discuss the major metabolic differences between human sepsis and the immune and metabolic characteristics of mouse models of sepsis so that the data that is currently in the manuscript can be appropriately interpreted regarding its significance and medical implications;
2. A constant throughout the manuscript is the message that increased pyruvate metabolism, and consequent rise in lactate concentrations, is detrimental. This is simplistic and contradicted by a vast body of literature. In fact, the authors encounter examples of this. It remains unclear whether preventing lactate production is desirable. As in many cases, the fact that a molecule correlates with severity or mortality does not necessarily imply that blocking or decreasing its concentration would be beneficial. It can be a compensatory response. While very high lactate concentrations can certainly cause organ pathology in sepsis, as the authors have previously demonstrated in mice, excessive lactate clearance is likely also detrimental. For several organs, lactate is actually protective! There is much clinical data to support this possibility. Discussing the consequences and effects of lactate-increased concentrations in a more mechanistic and nuanced manner would make the manuscript much more valuable and clinically accurate;
3. In a related point, the authors largely ignore the literature on the molecular effects of lactate, including on the regulation of the immune response and tissue protection. This should be corrected. <https://doi.org/10.1016/j.molcel.2023.09.034> is one recent example, but there are plenty of others like [10.1038/s41586-019-1678-1](https://doi.org/10.1038/s41586-019-1678-1) / [10.1126/sciadv.abi8602](https://doi.org/10.1126/sciadv.abi8602) / [10.15252/embr.202254685](https://doi.org/10.15252/embr.202254685). This should then feed on the metabolic reprogramming implications in sepsis.

Referee #3 (Remarks for Author):

Dear Author,

I have conducted a thorough review of the manuscript titled "Sepsis-induced changes in pyruvate metabolism: insights and therapeutic approaches" submitted to EMBO Molecular Medicine. This study, focusing on the metabolic perturbations associated with sepsis, particularly on pyruvate metabolism, presents a compelling area of research. However, the manuscript devotes too much space to the intrinsic functions of pyruvate-related enzymes. Instead, it should focus more on how each enzyme is regulated during sepsis, the metabolic changes induced by targeting these enzymes, and the consequences of these changes. While the manuscript repeatedly mentions that metabolic alterations occur, it lacks specific details on the nature of these changes. Additionally, while the outcomes of inhibiting or activating specific enzymes are discussed, there is insufficient consideration of the underlying mechanisms driving these results.

To strengthen the manuscript, several points identified require the author's attention, necessitating major revisions. The points are listed in the order they appear in the manuscript for clarity and ease of reference.

Major points:

1. (Page 13~14) The Role of MPC in Sepsis:

The role of MPC in sepsis is not definitively established. For instance, in sepsis caused by SARS-CoV-2 infection, MPC deletion in immune cells can increase the MDSC portion, suppress immunity, improve sepsis survival, and reduce weight loss (Inhibition of the mitochondrial pyruvate carrier simultaneously mitigates hyperinflammation and hyperglycemia in COVID-19, DOI: 10.1126/sciimmunol.adf0348a). It is recommended to discuss how the source of infection and the target cell of MPC influence the outcomes, rather than concluding that MPC is uniformly detrimental in all types of sepsis. This nuanced approach will provide a more comprehensive understanding of MPC's role in different sepsis contexts.

2. (Page 18~21) Insufficient Linkage Between PC-Mediated Metabolic Changes and Inflammation:

The text mentions the importance of PC in intermediary metabolism and its regulation by acetyl-CoA but does not sufficiently link these metabolic changes to inflammation. Include detailed explanations of how the metabolic changes mediated by PC, such as alterations in oxaloacetate levels, impact inflammatory pathways, ROS production, or other cellular responses relevant to sepsis.

3. Lack of Specificity in Describing Metabolic Pathways and Ambiguous Statements on Metabolic Shifts:

The manuscript describes pyruvate metabolism in broad terms, discussing concepts like metabolic shifts and changes in enzyme expression or phosphorylation patterns. However, it does not provide specific details on how these enzyme changes lead to alterations in metabolite levels and how these metabolic changes, in turn, result in inflammation, cell death, or other outcomes relevant to sepsis. Additionally, the manuscript mentions shifts in metabolic pathways in septic patients but does not provide sufficient data or references to support these claims. Including more detailed descriptions, specific examples, and supporting data or references would enhance the understanding of how altered enzyme activities impact sepsis outcomes. This would provide a clearer link between metabolic changes and the clinical manifestations of sepsis.

4. Insufficient Integration of Clinical Data:

The manuscript discusses several metabolic pathways and therapeutic strategies without adequately considering their clinical applicability. Additionally, it does not integrate clinical data effectively to support its claims about metabolic interventions in sepsis. Discussing the current state of therapeutic development based on these metabolic insights would enhance understanding of how these changes translate into treatments and their impact on patient outcomes. Given the preclinical nature of the results, it may be advisable to adjust the title to better reflect the scope, such as emphasizing "mechanisms and potential therapeutic targets" rather than "therapeutic approaches."

Minor points:

5. (Page 2) Inaccurate Statement in the abstract

The abstract states, "In mammals, pyruvate is the only substrate for lactate production," which seems too strong. It might be more precise to state that while pyruvate is the primary substrate for lactate production, other metabolic intermediates like alanine can be converted to pyruvate and subsequently to lactate. This would better reflect the metabolic flexibility and integration of pathways in cellular energy metabolism.

6. (Page 3) Misleading Statement on Starvation Response:

The manuscript states that septic patients trigger a starvation response due to an energy deficit but does not provide sufficient evidence or explanation for this claim. It is recommended to provide more context and references to support the claim that a starvation response is activated in septic patients due to energy deficits, or to discuss other factors that may contribute to the energy deficit. Additionally, more explanation is needed on the sepsis-associated failing starvation process to clarify its role and implications in the context of sepsis. This added detail would enhance the understanding of metabolic responses in septic patients.

7. (Page 8) Insufficient Detail on Enzyme Activities (Page 8)

The discussion on the role of various enzymes, such as PKM2 and LDHA, in sepsis is not detailed enough. Include more detailed explanations of how these enzymes are regulated in sepsis and their specific contributions to metabolic dysfunctions.

8. (Page 15) Clarification on PDH Complex Terminology:

The manuscript describes "PDH" as a large multi-component enzymatic complex comprising various catalytic, binding, and regulatory components. However, it would be more accurate to refer to this as the "PDH complex" (PDC) rather than simply "PDH." For example, on page 15, the first line should state "PDH complex is a large multi-component enzymatic complex (10

MDa) comprising three catalytic, binding, and regulatory components" instead of "PDH is a large multi-component enzymatic complex." It is important to consistently and clearly use the terms "PDH" and "PDH complex (PDC)" throughout the manuscript to avoid confusion. This clarification will help ensure precise communication of the enzymatic components and their functions.

9. (Page 16) Comprehensive Discussion on PDK Isoform Activities:

Based on the review of the Sugden & Holness (2006) reference, it is evident that PDK4 exhibits higher activity towards phosphorylation at site 2 compared with PDK1, PDK2, and PDK3. To provide a more comprehensive understanding of the regulation of PDH, I recommend including a discussion on the distinct phosphorylation site preferences and activities of all four PDK isoforms, with a mention of the significant role of PDK4. This addition would enrich the analysis by illustrating how the differential regulation by PDK isoforms influences PDH activity and, consequently, affects cellular energy metabolism in various physiological and pathological contexts.

10. (Page 22) Redundant Information

There is redundant information about the role of PDK and thiamine in pyruvate metabolism and sepsis. Consolidate this information to avoid repetition and improve the flow of the manuscript.

11. (Page 26) Oversimplified Statement on Pyruvate Metabolism:

The statement "The formed cytosolic pyruvate will be directly metabolized into lactate" could be seen as oversimplifying the process. It would be more accurate to state that "the formed cytosolic pyruvate is predominantly converted into lactate by lactate dehydrogenase," acknowledging that not all pyruvate is converted to lactate.

12. Clarification on Abbreviation :

The manuscript uses the abbreviation "GNEO," which seems to stand for "gluconeogenesis," but there is no information provided about this abbreviation. It is recommended to clearly define "GNEO" as "gluconeogenesis" upon its first use to avoid any confusion and ensure clarity for the readers. This clarification will help in understanding the context and maintaining consistency throughout the manuscript.

13. (page 26) Omission of Key Pathways:

The description omits the role of pyruvate in fatty acid synthesis. Pyruvate can be converted to acetyl-CoA, which can be used for fatty acid synthesis in the cytosol after being exported out of the mitochondria as citrate and then converted back to acetyl-CoA in the cytosol. Additionally, the conversion of pyruvate to malate by malic enzyme produces NADPH, which is crucial for steroid synthesis and other anabolic processes. Including these pathways would provide a more comprehensive overview of pyruvate metabolism, highlighting its role in lipid biosynthesis and related anabolic routes.

14. (Page 27) Clarification on Glutamate and the Urea Cycle:

The statement "The formed glutamate enters the urea cycle to eliminate its amino group" should be clarified. The amino group from glutamate is removed as ammonia, which then enters the urea cycle. The direct entry of glutamate into the urea cycle is not accurate; it is the ammonia derived from glutamate that is converted to urea. Including this clarification would provide a more precise explanation of the metabolic pathway.

Thank you for the opportunity to review this manuscript. I look forward to seeing the revised version.

Point-by-point response to the comments of the reviewers**Referee #1 (Remarks for Author):**

This paper overviews the cytosolic and mitochondrial enzymes involved in pyruvate metabolism and how their activities are disrupted in sepsis. Therapeutic strategies targeting these pyruvate-related enzymes in sepsis are also discussed.

General Comments:

This review nicely overviews the pathways involved in pyruvate metabolism. The concept is that pyruvate is the only source of lactate in mammals, and that accumulation of lactate is a biomarker of sepsis severity. The paper provides a comprehensive description of the different pathways involved in pyruvate metabolism, what happens to these pathways in sepsis, and what is known about targeting these pyruvate metabolic pathways to treat sepsis. However, I believe the authors need to better delineate how important modifying the pyruvate metabolic pathways are in modifying lactate levels, versus altering other metabolic and inflammatory processes.

General note: Additional text and adaptations are indicated in red in the manuscript and provided in italic below for clarity/to aid the reviewer.

Answer: We thank the reviewer for the positive comments. We do agree with the reviewer and we adapted the text to provide a better link between changes in pyruvate related enzymes and altered metabolic and inflammatory processes.

- We have included a comprehensive section, *Box 1: Metabolic alterations and failing starvation response in sepsis*, which provides a detailed explanation (with additional data and references) of the metabolic shifts observed in septic patients for more clarity:

“Box 1: Metabolic alterations & failing starvation response in sepsis”

Hallmarks of sepsis include immune activation, phagocytosis, acute phase reactant production, fever, tachycardia, and tachypnea, which all require increased supraphysiological energy supplies (Van Wyngene et al, 2018; Wasyluk & Zwolak, 2021). Despite this, septic patients are often unwilling or unable to consume food and exhibit mitochondrial dysfunction, resulting in a reduced capacity of cells to produce ATP (Peterson et al, 2010; Wang et al, 2016; Singer & Brealey, 1999). Muscle biopsies from septic patients are characterized by a diminished ATP/ADP ratio, correlated with sepsis-induced multiple organ failure and poor outcome (Brealey et al, 2002; Fredriksson et al, 2006). This energy imbalance in septic individuals leads to the activation of a starvation response, as evidenced by the following observations. (1) Adipose tissue in septic patients shows increased lipolysis, resulting in higher blood levels of free fatty acids (FFAs), triglycerides, and glycerol, which are significantly higher in septic non-survivors compared to sepsis survivors (Ilias et al, 2014; Rittig et al, 2016; Langley et al, 2013; Lee et al, 2015; Wang et al, 2020). (2) Hepatic cellular glycogen reserves are heavily depleted (Vandewalle et al, 2021). (3) Skeletal muscle proteolysis is activated, leading to increased concentrations of amino acids (e.g. alanine and glutamine) in the blood (Long et al, 1981; Su et al, 2015; Langley et al, 2013; Wang et al, 2020). Interestingly, many studies show that the conversion of these energy-rich substrates into useful metabolites (e.g. acetyl-CoA, ketone bodies and glucose) is disturbed in septic individuals. A proteomic and metabolomic study on plasma samples of sepsis patients shows decreased protein levels of nine fatty acid transporters but elevated protein levels of two fatty acid-binding proteins in sepsis non-survivors, suggesting a profound defect in fatty acid β -oxidation (Langley et al, 2013, 2014). This defect is caused by impaired functioning of the peroxisome proliferator-

activated α receptor (PPAR α) as septic individuals exhibit reduced expression of PPAR α and PPAR α -dependent genes correlating with severity (Wong et al, 2009; Standage et al, 2012; Van Wyngene et al, 2020). This results in harmful FFA accumulation leading to lipotoxicity and contributing to multiple organ damage and failure (Van Wyngene et al, 2020). On the other hand, septic non-survivors show elevated plasma levels of citrate, malate, pyruvate, dihydroxacetone, lactate and gluconeogenic amino acids, suggesting alterations in glycolysis, the TCA cycle and GNEO pathways (Langley et al, 2013, 2014; Wang et al, 2020). Indeed, studies show that GNEO is critical for surviving sepsis but often fails due to several factors. Firstly, the glucocorticoid receptor, which regulates the transcription of genes encoding GNEO enzymes like Pck1 (encoding PEPCCK) and G6Pc (encoding G6Pase), becomes dysfunctional in sepsis, leading to impaired GNEO (Vandewalle et al, 2021). Secondly, the production of pro-inflammatory cytokines following exposure to endotoxic shock decreases the transcription of the rate-limiting enzyme Pck1 by reducing the expression of the nuclear receptor cofactor PGC1 α , the latter being an essential cofactor for Pck1 transcription (Chichelnitskiy et al, 2009). Lastly, oxidative inhibition of liver G6Pase driven by iron further contributes to GNEO failure during sepsis. Interestingly, counteracting G6Pase repression using ferritin has been shown to preserve GNEO and reduce sepsis mortality, highlighting the importance of sustaining this process during sepsis (Weis et al, 2017). Altogether, this suggests the presence of a sepsis-associated failing starvation response correlating with a bad outcome.”

- Additionally, we have adapted the text wherever possible to highlight more how specific changes in enzyme activity result in inflammation, cell death, or other outcomes relevant to sepsis, see below:

Pyruvate kinase:

“In sepsis, metabolic reprogramming from OXPHOS to aerobic glycolysis is a typical hallmark of activated immune cells in order to accommodate their high energy requirements for performing their inflammatory responses. Indeed, serum PKM2 levels are increased in sepsis patients and are positively correlated with blood glucose, lactate, LDH and disease severity (Wang et al, 2023). Additionally, urinary PKM2 expression is elevated in patients with sepsis-associated acute kidney injury and positively correlates with serum creatinine levels (Jiajun et al, 2024). Hence, this suggests that septic patients are characterized by a PKM2-mediated glucose metabolic reprogramming, and that PKM2 could potentially be used as a prognostic biomarker. Generally, PKM2 can be induced by hypoxia or after LPS stimulation (Wasyluk & Zwolak, 2021). More specifically, LPS-stimulated macrophages show increased PKM2 expression and simultaneous phosphorylation of PKM2 on Tyrosine 105 keeping PKM2 in its dimeric, inactive form. With this, PKM2 can translocate to the nucleus resulting in the activation of hypoxia-inducible factor 1 α (HIF1 α) via the formation of a PKM2-HIF1 α complex (Palsson-Mcdermott et al, 2015). On the one hand, PKM2-mediated activation of HIF1 α leads to a metabolic shift towards aerobic glycolysis by inducing the expression of glycolysis-related genes, resulting in elevated lactate production. In turn, lactate inhibits histone deacetylase activity, resulting in high-mobility box 1 (HMGB1) hyperacetylation and its subsequent release (Yang et al, 2014). Additionally, this PKM2-mediated lactate production induces EIF2AK2 phosphorylation resulting in NLRP3 and AIM2 inflammasome activation and subsequent pro-inflammatory mediator release (IL-1 β , IL-18 and HMGB1) in LPS-treated macrophages (Xie et al, 2016). On the other hand, PKM2-HIF1 α complex can bind to the IL-1 β promoter and induces excessive IL-1 β production in LPS-activated macrophages (Palsson-Mcdermott et al, 2015). Furthermore, these studies indicate that the potential PKM2 inhibitor, shikonin,

reduces PKM2 activity in LPS stimulated macrophages. This reduction protects mice from lethal endotoxemia and sepsis by partially decreasing lactate production, and consequently pro-inflammatory cytokines release (Yang et al, 2014; Xie et al, 2016). This study confirms the need for therapeutic interventions addressing PKM2-mediated aerobic glycolysis as a metabolic control in inflammation for the treatment of sepsis .”

“PKM2-mediated protection against SIC is also necessary to attenuate LPS-induced mitochondrial damage marked by diminished ATP production, reduced mitochondrial respiratory complex I/III activities and increased ROS production, eventually resulting in enhanced myocardial inflammation and impaired cardiac function. This is mediated via PKM2-dependent phosphorylation of prohibitin 2 (essential for the preservation of mitochondrial function and structure) in order to limit LPS-mediated prohibitin 2 degradation (Ren et al, 2024; Du et al, 2024). Hence, cardiomyocyte-specific PKM2 overexpression exhibits protective effects against SIC in mice with LPS or gram-negative bacteria induced sepsis (Ni et al, 2022; Du et al, 2024).”

Lactate dehydrogenase:

To provide more clarity regarding the effects of lactate, an additional text in the manuscript was provided (see below) along with an additional section: Box 2: *The versatile role of lactate in sepsis*, see below (comment 1).

“As mentioned in the previous section, sepsis is associated with glucose metabolic reprogramming of immune cells (i.e. Warburg phenotype) potentially mediated via elevated HIF1 α activation. Apart from PKM2 and hypoxic conditions, bacterial products (e.g. LPS via mTOR activation) and pro-inflammatory cytokines (e.g. TNF α) can induce HIF1 α activity under normoxic conditions (Nishi et al, 2008; Rgueira et al, 2009). In turn, HIF1 α induces the expression of glycolysis-related genes including LDH, thereby increasing the aerobic glycolytic flux responsible for elevated pyruvate levels driving LDH activity in producing lactate (Garcia-Alvarez et al, 2014; TAN et al, 2021). Indeed, several clinical studies have provided evidence that serum lactate levels are elevated and may hold significance in predicting mortality in patients with septic shock (defined by Sepsis-3) (Ryoo et al, 2018; Lee et al, 2021). Moreover, sepsis patients show increased serum LDH levels which is positively correlated with serum lactate, IL-1 β and 28-day mortality, thereby suggesting that glucose metabolic reprogramming of immune cells contributes to sepsis mortality (Lu et al, 2018; Frenkel et al, 2023). Moreover, many in vitro studies show increased levels of LDHA upon LPS stimulation in macrophages (Yang et al, 2014; Palsson-Mcdermott et al, 2015; Xie et al, 2016; Zhang et al, 2022)”

“This protective effect results from the role of lactate in mediating HMGB1 lactylation via its direct uptake through monocarboxylate transporters (MCTs) or via a p300/CBP dependent mechanism in activated macrophages. Lactate can also bind to its receptor, G-protein coupled receptor 81 (GPR81), leading to decreased expression of SIRT1 deacetylase, thereby inducing increased HMGB1 acetylation in activated macrophages. Eventually, lactylation and acetylation of HMGB1 lead to its exosomal secretion in macrophages, contributing to endothelial barrier dysfunction and vascular permeability which exacerbates sepsis progression”

Pyruvate carboxylase

We have incorporated studies that demonstrate how PC-mediated metabolic changes have an impact on inflammatory-induced NO synthesis, glutathione antioxidant mechanism, inflammation-induced ROS accumulation and the viral innate immune response, see below.

“Interestingly, pancreatic islet β -cells also exhibit reduced PC activity when treated with pro-inflammatory cytokines (TNF α , IL-1 β and IFN- γ). Increasing PC activity, and thus PC-mediated TCA anaplerosis, protects β -cells against inflammation toxicity by increasing aspartate availability through oxaloacetate conversion via aspartate aminotransferase. Elevated aspartate creates an argininosuccinate shunt, activating the urea cycle and redirecting arginine metabolism towards ureagenesis. This limits arginine use for inflammation-induced NO synthesis, reducing oxidative damage and preventing cell death of pancreatic islet β -cells (Fu et al, 2020). PC activity can also facilitate an extra protective mechanism against oxidative stress by controlling the NADPH/NADP⁺ ratio through sustaining malic enzyme activity. Liver-specific PC knockout (LPCKO) mice show diminished malate production and accumulation of pyruvate, resulting in altered malic enzyme activity leading to NADPH/NADP⁺ depletion, diminished antioxidant defenses via a reduced glutathione antioxidant mechanism, and increased liver inflammation susceptibility (Cappel et al, 2019). In addition to preserving the glutathione antioxidant activity, PC is crucial for de novo glutathione synthesis, thereby limiting inflammation-induced ROS accumulation in pancreatic islet β -cells during inflammatory stress (Fu et al, 2021). Conversely, a recent study by Liang et al. demonstrated increased PC activity in in vitro (LPS-treated colorectal cells and LPS-treated monocytes) and in vivo (Dextran sulfate sodium (DSS)-induced) models of colitis. This was accompanied with elevated NF- κ B signaling, increased production of inflammatory cytokines and ROS, decreased pyruvate content, and accumulation of acetyl-CoA, oxaloacetate and lactate. Moreover, treatment with anemoside B4, a specific PC inhibitor, could ameliorate the inflammatory response, oxidative stress, and could regulate the levels of pyruvate and its downstream metabolites. These findings suggest that PC-mediated changes in pyruvate and acetyl-CoA levels are essential in mediating the inflammatory response and ROS production in LPS-treated colorectal cells. Hence, targeting PC with anemoside B4 might be a potential anti-inflammatory approach (Liang et al, 2024). Notably, PC activity is also crucial in viral innate immune responses by targeting RIG-I-MAVS-TRAF6-signaling pathway. This promotes NF- κ B activation and increases production of interferons and pro-inflammatory cytokines in viral infected cells (Cao et al, 2016).”

Alanine aminotransferase, mitochondrial pyruvate carrier and pyruvate dehydrogenase:

As research towards the role of ALT and MPC in sepsis individuals is currently limited, we could not provide additional insights or references regarding the role of ALT or MPC-induced changes in metabolite levels and inflammation. Additionally, the effects of PDC activity are already extensively described in the manuscript and due to word limit constrains and the risk of overloading the manuscript, no additional insights could be provided. We believe that the current level of detail is sufficient to convey the critical role of these enzymes in the context of sepsis while maintaining the overall clarity and focus of the manuscript.

In sepsis, it is not only the accumulation of lactate that is harmful, but also the development of acidosis that results from glycolysis uncoupled from pyruvate oxidation or carboxylation, versus pyruvate destined for lactate production. This concept is largely ignored in this review, and I believe needs to be addressed.

Answer: We agree and we have added some key insights into the cause of lactic acidosis during sepsis in an additional section from Box 2: *“The versatile role of lactate in sepsis”*, discussing the role of lactate in this condition, see paragraph below.

“Of note, sepsis is frequently associated with lactic acidosis, which is typically defined as a blood lactate concentration exceeding 5 mM combined with a blood pH below 7.35 (Mizock, 1992). The precise role of glycolysis in inducing lactatemia or lactic acidosis remains unclear. During glycolysis, one molecule of glucose yields two molecules of pyruvate, 2 ATP, 2 NADH, and 2 protons (H⁺). When pyruvate is converted to lactate by LDH, NADH is reverted to NAD⁺, consuming an equivalent amount of protons. Thus, an increase in lactate production leading to hyperlactatemia is not inherently acidifying. The primary source of protons, and hence acid, is ATP hydrolysis. The Krebs cycle actively consumes these protons, so a reduction in Krebs cycle flux due to impaired oxygen utilization or mitochondrial dysfunction leads to acid accumulation. Simultaneously, reduced Krebs cycle activity results in lactate production. The cotransport of lactate and protons by MCT causes tissue hypoxia-related acidosis to manifest as “lactic acidosis” clinically. Therefore, the term “lactic acidosis” is a misnomer, and “lactate-associated acidosis” is more accurate, as lactate itself does not cause acidosis (Müller et al, 2023). In sepsis, this lactate-associated acidosis is primarily compensated for by the kidneys, which decrease strong anions, thus widening the strong ion difference. Consequently, the severity of acidemia heavily depends on renal function (Gattinoni et al, 2019).”

Specific Comments:

- 1) While a detailed discussion is provided on lactate production from LDH in sepsis, there is almost no discussion of the fate of the lactate. Recent studies have suggested that inhibition of the monocarboxylic acid transporter (MCT) can favourably affect pyruvate fate in muscle. I believe some discussion of the potential of MCT inhibition in sepsis is warranted.

Answer: We agree with the reviewer and have added an additional paragraph in Box 2: *“The versatile role of lactate in sepsis”* discussing the fate of lactate and the effect of MCT inhibition during sepsis. See below.

“For decades, lactate was considered as an inert by-product of glycolysis rather than a bioactive molecule. Interestingly, lactate can also function as a signaling molecule and can be transported into cells by the monocarboxylate transporters (MCTs), which play a crucial role in metabolic communication between cells. MCT4 is specialized for lactate export, while MCT1 can either import or export lactate, depending on whether it is expressed by oxidative or glycolytic cells.

Lactate is often considered harmful in sepsis; Injection in septic animals worsens sepsis, whereas inhibiting lactate production with the LDH inhibitor, oxamate, improves sepsis survival (Yang et al, 2022). In macrophages, extracellular lactate uptake leads to HMGB1 lactylation, promoting its release and increasing endothelial permeability (Yang et al, 2022). Inhibiting lactate uptake with CHC, an MCT inhibitor, suppresses lactate-induced HMGB1 lactylation in macrophages. In neutrophils, lactate uptake through MCT1 upregulates PD-L1 expression, delaying apoptosis in these cells. Administration of the selective MCT1 inhibitor AZD3965 increases neutrophil apoptosis leading to enhanced survival in CLP mice (Fei et al, 2024). Research by Vandewalle et al. even demonstrates that lactate significantly contributes to sepsis lethality as hyperlactatemia, in combination with a sepsis-induced glucocorticoid

resistance, results in a lethal vascular collapse via uncontrolled vascular endothelial barrier dysfunction (Vandewalle et al, 2021).

Contrary to these findings, other researchers have reported that lactate infusion can improve sepsis outcomes by enhancing hemodynamics and reducing inflammation (Walenta et al, 2000). Lactate's role in immunosuppression is well-documented in cancer research and also in inflammatory conditions, where it has been shown to reduce organ damage (Walenta et al, 2000; Hoque et al, 2014). Recently, lactate produced by monocyte-derived human tolerogenic dendritic cells has been found to decrease T cell proliferation, delaying graft-versus-host disease (Marin et al, 2019).

Lactate also has direct protective effects on organs such as the heart and brain. In the heart, it serves as an energy source, and pharmacological inhibition of MCT4—which blocks lactate export—mitigates heart failure in mice. Systemic lactate deprivation using dichloroacetate and ICI-118551 (inhibiting respectively PDK and β_2 -adrenergic receptors) is linked to decreased myocardial energetics, reduced cardiovascular performance, and early death in endotoxic shock (Levy et al, 2007). In the brain, lactate compensates for reduced glucose uptake following traumatic brain injury (Glenn et al, 2015). Given the starvation response observed in sepsis patients, it stands to reason that lactate is necessary for maintaining myocardial and cerebral metabolism during sepsis (Vandewalle & Libert, 2022).

*Additionally, lactate can directly affect pathogens. For example, *Candida albicans* utilizes lactate to increase its resistance to host-relevant stressors, enhance biofilm formation, and evade macrophage recognition (Williams & Lorenz). Conversely, lactate inhibits the growth of *Salmonella* in vitro and reduces mortality from *Salmonella* infection in vivo (Iraporda et al, 2017).*

In summary, lactate acts as a double-edged sword, with its effects varying based on dose, type (lactic acid versus lactate anion), metabolic state, cell type, and pathology. While high lactate levels are clearly detrimental in sepsis, some clinicians now recognize that mild lactate production during sepsis is a compensatory mechanism. The so-called "Lactate Shuttle Theory" leverages lactate as a signaling and energy molecule to meet increased energy demands and dampen the inflammatory response during sepsis (Brooks, 2018)."

- 2) Pyruvate dehydrogenase (PDH) activity is markedly affected by fatty acids, which can increase dramatically in sepsis. The potential role of increased fatty acid oxidation in inhibiting PDH in sepsis should be discussed.

Answer: We appreciate the reviewer's insightful comment. Indeed, studies show that fatty acids can inhibit PDC activity due to increased fatty acid oxidation leading to increased levels of NADH and acetyl-CoA. This results in the activation of PDKs leading to the phosphorylation, and consequently inactivation of the PDC enzyme (Batenburg & Olson, 1976; Zhou & Grill, 1995; Zhou et al, 1996; Bradley et al, 2008). However, in the context of sepsis, it is generally known that fatty acid oxidation is impaired resulting in reduced production of NADH and acetyl-CoA (Van Wyngene et al, 2020). So, this cannot be the reason for impaired PDC activity in sepsis. We have therefore decided not to mention this in the manuscript.

- 3) I am presuming that "GNEO" stands for gluconeogenesis. This could be more clearly indicated in the manuscript.

Answer: Thank you for pointing this out. Indeed, “GNEO” stands for gluconeogenesis. We have now revised the text to clearly indicate that GNEO refers to gluconeogenesis, and we have also added this abbreviation to the gluconeogenesis definition in the glossary.

- 4) It is stated that PK is the rate-limiting enzyme of glycolysis. This is not universally accepted, and many studies propose phosphofructokinase is the rate-limiting enzyme for glycolysis.

Answer: We thank the reviewer for this comment. We have adapted the text for more clarification. See below.

“PK catalyzes the final step of glycolysis, involving the irreversible transfer of a phospho-group from PEP to ADP, yielding a molecule of pyruvate and ATP. It is one of the rate-limiting enzymes of glycolysis, alongside hexokinase and phosphofructokinase, and is therefore of crucial importance as a key metabolic control point”

- 5) I believe the role of PEPCCK in gluconeogenesis in sepsis is worth discussing.

Answer: In accordance to the comment in point 7, we have described the role of PEPCCK (Pck1) and gluconeogenesis more in detail in an additional paragraph on the starvation response (Box1: *Metabolic alterations and failing starvation response in sepsis*), see below (point 7). Additionally, PEPCCK has been included in Figure 2 (“The metabolic fates of pyruvate”) to provide a more detailed overview.

- 6) It is stated that "Unfortunately, research towards PC activity in sepsis is currently constrained." What does this mean?

Answer: We understand this comment of the reviewer and that “constrained” is not the ideal wording. We adapted the manuscript for more clarity, see below.

“Unfortunately, there is limited research into the role of PC activity in sepsis.”

- 7) It is not clear how important gluconeogenesis is in sepsis, and whether inhibiting or stimulating gluconeogenesis would be desirable.

Answer: We have made this clearer with an extra paragraph (Box 1: *Metabolic alterations and failing starvation response in sepsis*), see below.

“Indeed, studies show that GNEO is critical for surviving sepsis but often fails due to several factors. Firstly, the glucocorticoid receptor, which regulates the transcription of genes encoding GNEO enzymes like Pck1 (encoding PEPCCK) and G6Pc (encoding G6Pase), becomes dysfunctional in sepsis, leading to impaired GNEO (Vandewalle et al, 2021). Secondly, the production of pro-inflammatory cytokines following exposure to endotoxic shock decreases the transcription of the rate-limiting enzyme Pck1 by reducing the expression of the nuclear receptor cofactor PGC1a, the latter being an essential cofactor for Pck1 transcription (Chichelnitskiy et al, 2009). Lastly, oxidative inhibition of liver G6Pase driven by iron further contributes to GNEO failure during sepsis. Interestingly, counteracting G6Pase repression using ferritin has been shown to preserve GNEO and reduce sepsis mortality, highlighting the importance of sustaining this process during sepsis (Weis et al, 2017).”

Referee #2 (Remarks for Author):

Nuyttens et al. review sepsis-induced changes in pyruvate metabolism and their therapeutic implications. This is an in-depth and authoritative manuscript. Here are a few major points that, in this reviewer’s opinion, are critical to improving the breath, appeal, and relevance of this work:

General note: Additional text and adaptations are indicated in red in the manuscript and provided in italic below for clarity/to aid the reviewer.

1. The review is centered on Sepsis, which is a condition that is so relevant because it is a leading cause of human mortality and disability, accounting for ~20% of all global human mortality. However, the authors chose mouse models that have severe limitations to satisfactorily model the condition as the focus of the review, only citing human data that supports mouse observations. It would be more significant to do the exact opposite: review human data and cite mouse data when that has mechanistic value to understand the observations. As done here, it runs into the problem that often very promising and exciting observations do not translate into progress in therapeutic interventions. A good example in the current work is DCA, which has shown disappointing results in clinical trials already several decades ago. In this context, it would be useful that the authors describe and discuss the major metabolic differences between human sepsis and the immune and metabolic characteristics of mouse models of sepsis so that the data that is currently in the manuscript can be appropriately interpreted regarding its significance and medical implications;

Answer: This is a good comment of the reviewer. We have revised the text to emphasize human data wherever possible. However, it should be noted that clinical research in sepsis patients poses significant challenges and is restricted. In addition, current research on the roles of ALT, MPC and PC in sepsis models is notably limited, and to our knowledge, there are no clinical observations available for these enzymes in sepsis patients. We have therefore highlighted the need for more clinical sepsis research in the “Pending Issues” section. Despite these limitations, we have focused to cite human data regarding changes in PK, LDH and PDH as these enzymes are more commonly studied in sepsis research. Furthermore, due to word limit constraints and to stay within the scope of this review, it was not possible to include a discussion regarding the major differences between human sepsis and mouse model of sepsis.

Please see the revised sections below:

- **Pyruvate kinase:** We have cited human data (see below) showing increased serum PKM2 levels that correlate with glucose, lactate, LDH and disease severity. This is followed by mouse data that provide mechanistic insights into PKM2-mediated aerobic glycolysis, its consequences and potential therapeutic strategies

“In sepsis, metabolic reprogramming from OXPHOS to aerobic glycolysis is a typical hallmark of activated immune cells in order to accommodate their high energy requirements for performing their inflammatory responses. Indeed, serum PKM2 levels are increased in sepsis patients and are positively correlated with blood glucose, lactate, LDH and disease severity (Wang et al, 2023). Additionally, urinary PKM2 expression is elevated in patients with sepsis-associated acute kidney injury, positively correlated with serum creatinine levels (Jiajun et al, 2024). Hence, this suggests that septic patients are characterized by a PKM2-mediated glucose metabolic reprogramming, and that PKM2 could potentially be used as a prognostic biomarker.”

- **Lactate dehydrogenase:** We have added additional human data, along with *in vitro* studies, showing increased serum LDH levels correlated with higher lactate, IL-1b and mortality as evidence for increased LDH activity in septic subjects. Additionally, increased levels of lactate in the skeletal muscle of septic shock patients are observed by Levy *et al.* indicating increased

LDHA activity in the skeletal muscle of sepsis patients. After this, the protection mechanism by therapeutically targeting LDHA activity in sepsis models is discussed.

“Indeed, several clinical studies have provided evidence that serum lactate levels are elevated and may hold significance in predicting mortality in patients with septic shock (defined by Sepsis-3) (Ryoo et al, 2018; Lee et al, 2021). Moreover, sepsis patients show increased serum LDH levels which is positively correlated with serum lactate, IL-1 β and 28-day mortality, thereby suggesting that glucose metabolic reprogramming of immune cells contributes to sepsis mortality (Lu et al, 2018; Frenkel et al, 2023). Moreover, many in vitro studies show increased levels of LDHA upon LPS stimulation in macrophages (Yang et al, 2014; Palsson-Mcdermott et al, 2015; Xie et al, 2016; Zhang et al, 2022). Apart from lactate production by immune cells, Levy et al. observe elevated levels of pyruvate and lactate in the skeletal muscle of septic shock patients. Interestingly, this is driven by the release of epinephrine which triggers Na⁺K⁺ ATPase activity to induce membrane hyperpolarization. This increased activity of the Na⁺K⁺ pump necessitates greater ATP consumption, which is facilitated by an enhanced aerobic glycolysis potential, and thus increased LDH activity. Consequently, this process is associated with Na⁺K⁺ ATPase-linked lactate production in skeletal muscle, as evidenced by the administration of a specific Na⁺K⁺ ATPase inhibitor, ouabain, which significantly reduces skeletal muscle pyruvate and lactate levels in patients with septic shock (Levy et al, 2005). Altogether, this highlights the need for therapeutic interventions targeting LDHA in sepsis.”

- Pyruvate dehydrogenase: We have mentioned one clinical study regarding impaired PDC activity, as well as two DCA clinical trials that demonstrated reduced lactate levels with DCA administration but no survival benefit. However, despite the findings of Stacpoole *et al.*, many studies in murine sepsis models have shown improved survival with DCA treatment (Stacpoole et al, 1992). Given that only one clinical study reported no survival benefit and no other clinical studies with DCA have been performed, further clinical trials investigating DCA are still worth pursuing and may require optimization.

“Moreover, peripheral blood mononuclear cells of septic patients exhibit diminished PDC levels and activity compared to healthy controls which was associated with reduced survival chance (Nuzzo et al, 2015). Interestingly, targeting PDK with the prototypic drug, dichloroacetate (DCA, a structural analogue of pyruvate), lowers phosphorylation on the E1 α subunit which ultimately improves PDH activity and lowers blood lactate levels. Indeed, DCA treatment improves oxygen consumption, pyruvate oxidation and reduces plasma lactate concentration in septic patients (Stacpoole et al, 1992; Gore & Demaria, 1996). However, the clinical trial of Stacpoole et al. shows that DCA treatment fails to improve the survival rate (Stacpoole et al, 1992). Despite this, DCA protection has been widely observed in many studies using sepsis models.”

Additional text in the PDC conclusion paragraph:

“However, clinical trials studying the efficacy of DCA in septic patients remain limited. DCA treatment could effectively reduce hyperlactatemia, but fails to improve the survival rates (Stagpoole et al, 1992). Nevertheless, further exploration of the protective role of DCA in sepsis patients is strongly recommended.”

2. A constant throughout the manuscript is the message that increased pyruvate metabolism, and consequent rise in lactate concentrations, is detrimental. This is simplistic and contradicted by a vast body of literature. In fact, the authors encounter examples of this. It remains unclear whether preventing lactate production is desirable. As in many cases, the fact that a molecule correlates with severity or mortality does not necessarily imply that blocking or decreasing its concentration would be beneficial. It can be a compensatory response. While very high lactate concentrations can certainly cause organ pathology in sepsis, as the authors have previously demonstrated in mice, excessive lactate clearance is likely also detrimental. For several organs, lactate is actually protective! There is much clinical data to support this possibility. Discussing the consequences and effects of lactate-increased concentrations in a more mechanistic and nuanced manner would make the manuscript much more valuable and clinically accurate;

Answer: We agree with the reviewer and have added an additional section (Box2: *The versatile role of lactate in sepsis*) discussing the role of lactate in sepsis, see below:

“For decades, lactate was considered as an inert by-product of glycolysis rather than a bioactive molecule. Interestingly, lactate can also function as a signaling molecule and can be transported into cells by the monocarboxylate transporters (MCTs), which play a crucial role in metabolic communication between cells. MCT4 is specialized for lactate export, while MCT1 can either import or export lactate, depending on whether it is expressed by oxidative or glycolytic cells.

Lactate is often considered harmful in sepsis; Injection in septic animals worsens sepsis, whereas inhibiting lactate production with the LDH inhibitor, oxamate, improves sepsis survival (Yang et al, 2022). In macrophages, extracellular lactate uptake leads to HMGB1 lactylation, promoting its release and increasing endothelial permeability (Yang et al, 2022). Inhibiting lactate uptake with CHC, an MCT inhibitor, suppresses lactate-induced HMGB1 lactylation in macrophages. In neutrophils, lactate uptake through MCT1 upregulates PD-L1 expression, delaying apoptosis in these cells. Administration of the selective MCT1 inhibitor AZD3965 increases neutrophil apoptosis leading to enhanced survival in CLP mice (Fei et al, 2024). Research by Vandewalle et al. even demonstrates that lactate significantly contributes to sepsis lethality as hyperlactatemia, in combination with a sepsis-induced glucocorticoid resistance, results in a lethal vascular collapse via uncontrolled vascular endothelial barrier dysfunction (Vandewalle et al, 2021).

Contrary to these findings, other researchers have reported that lactate infusion can improve sepsis outcomes by enhancing hemodynamics and reducing inflammation (Walenta et al, 2000). Lactate's role in immunosuppression is well-documented in cancer research and also in inflammatory conditions, where it has been shown to reduce organ damage (Walenta et al, 2000; Hoque et al, 2014). Recently, lactate produced by monocyte-derived human tolerogenic dendritic cells has been found to decrease T cell proliferation, delaying graft-versus-host disease (Marin et al, 2019).

Lactate also has direct protective effects on organs such as the heart and brain. In the heart, it serves as an energy source, and pharmacological inhibition of MCT4—which blocks lactate export—mitigates heart failure in mice. Systemic lactate deprivation using dichloroacetate and ICI-118551 (inhibiting respectively PDK and β 2-adrenergic receptors) is linked to decreased myocardial energetics, reduced cardiovascular performance, and early death in endotoxic shock (Levy et al, 2007). In the brain, lactate compensates for reduced glucose uptake following traumatic brain injury (Glenn et al, 2015). Given the starvation response

observed in sepsis patients , it stands to reason that lactate is necessary for maintaining myocardial and cerebral metabolism during sepsis (Vandewalle & Libert, 2022).

Additionally, lactate can directly affect pathogens. For example, Candida albicans utilizes lactate to increase its resistance to host-relevant stressors, enhance biofilm formation, and evade macrophage recognition (Williams & Lorenz). Conversely, lactate inhibits the growth of Salmonella in vitro and reduces mortality from Salmonella infection in vivo (Iraporda et al, 2017).

In summary, lactate acts as a double-edged sword, with its effects varying based on dose, type (lactic acid versus lactate anion), metabolic state, cell type, and pathology. While high lactate levels are clearly detrimental in sepsis, some clinicians now recognize that mild lactate production during sepsis is a compensatory mechanism. The so-called “Lactate Shuttle Theory” leverages lactate as a signaling and energy molecule to meet increased energy demands and dampen the inflammatory response during sepsis (Brooks, 2018).

Of note, sepsis is frequently associated with lactic acidosis, which is typically defined as a blood lactate concentration exceeding 5 mM combined with a blood pH below 7.35. The precise role of glycolysis in inducing lactatemia or lactic acidosis remains unclear. During glycolysis, one molecule of glucose yields two molecules of pyruvate, 2 ATP, 2 NADH, and 2 protons (H⁺). When pyruvate is converted to lactate by LDH, NADH is reverted to NAD⁺, consuming an equivalent amount of protons. Thus, an increase in lactate production leading to hyperlactatemia is not inherently acidifying. The primary source of protons, and hence acid, is ATP hydrolysis. The Krebs cycle actively consumes these protons, so a reduction in Krebs cycle flux due to impaired oxygen utilization or mitochondrial dysfunction leads to acid accumulation. Simultaneously, reduced Krebs cycle activity results in lactate production. The cotransport of lactate and protons by MCT causes tissue hypoxia-related acidosis to manifest as “lactic acidosis” clinically. Therefore, the term “lactic acidosis” is a misnomer, and “lactate-associated acidosis ” is more accurate, as lactate itself does not cause acidosis (Müller et al, 2023). In sepsis, this lactate-associated acidosis is primarily compensated for by the kidneys, which decrease strong anions, thus widening the strong ion difference. Consequently, the severity of acidemia heavily depends on renal function (Gattinoni et al, 2019).”

3. in a related point, the authors largely ignore the literature on the molecular effects of lactate, including on the regulation of the immune response and tissue protection. This should be corrected. <https://doi.org/10.1016/j.molcel.2023.09.034> is one recent example, but there are plenty of others like [10.1038/s41586-019-1678-1](https://doi.org/10.1038/s41586-019-1678-1) / [10.1126/sciadv.abi8602](https://doi.org/10.1126/sciadv.abi8602) / [10.15252/embr.202254685](https://doi.org/10.15252/embr.202254685). This should then feed on the metabolic reprogramming implications in sepsis.

Answer: We appreciate the reviewer’s valuable suggestion. However, due to the word limit constraints of the manuscript, it is impossible to provide a complete discussion of the role of lactate in different immune cells with the references provided by the reviewer. Therefore, we have focused on the immunomodulating and tissue protective effects of lactate during sepsis or related diseases specifically (see additional paragraph in comment above; Box 2: *The versatile role of lactate in sepsis*) and also in Table 1 the role of lactate in different immune cells during sepsis is described.

Referee #3 (Remarks for Author):

Dear Author,

I have conducted a thorough review of the manuscript titled "Sepsis-induced changes in pyruvate metabolism: insights and therapeutic approaches" submitted to EMBO Molecular Medicine. This study, focusing on the metabolic perturbations associated with sepsis, particularly on pyruvate metabolism, presents a compelling area of research. However, the manuscript devotes too much space to the intrinsic functions of pyruvate-related enzymes. Instead, it should focus more on how each enzyme is regulated during sepsis, the metabolic changes induced by targeting these enzymes, and the consequences of these changes. While the manuscript repeatedly mentions that metabolic alterations occur, it lacks specific details on the nature of these changes. Additionally, while the outcomes of inhibiting or activating specific enzymes are discussed, there is insufficient consideration of the underlying mechanisms driving these results.

To strengthen the manuscript, several points identified require the author's attention, necessitating major revisions. The points are listed in the order they appear in the manuscript for clarity and ease of reference.

General note: Additional text and adaptations are indicated in red in the manuscript and provided in italic below for clarity/to aid the reviewer.

Major points:

1. (Page 13~14) The Role of MPC in Sepsis:

The role of MPC in sepsis is not definitively established. For instance, in sepsis caused by SARS-CoV-2 infection, MPC deletion in immune cells can increase the MDSC portion, suppress immunity, improve sepsis survival, and reduce weight loss (Inhibition of the mitochondrial pyruvate carrier simultaneously mitigates hyperinflammation and hyperglycemia in COVID-19, DOI: 10.1126/sciimmunol.adf0348a). It is recommended to discuss how the source of infection and the target cell of MPC influence the outcomes, rather than concluding that MPC is uniformly detrimental in all types of sepsis. This nuanced approach will provide a more comprehensive understanding of MPC's role in different sepsis contexts.

Answer: We want to thank the reviewer for this interesting additional insight. We have included the study of Zhu *et al.* in the manuscript and we have adapted the conclusion paragraph of MPC to emphasize that the role of MPC in inducing macrophage inflammatory responses is dependent on the type of infection and the specific-cell type, see below.

“Conversely, a recent study of Zhu et al. demonstrates that myeloid-specific Mpc deletion or pharmacological inhibition of MPC with MSDC-0602K (MSDC) mitigates disease severity after influenza or SARS-CoV-2-induced pneumonia. Specifically, MSDC administration targets lung macrophages (and not BMDM) leading to a suppressed pulmonary hyperinflammatory response via increased mitochondrial fitness resulting in reduced levels of HIF1 α -stabilizing metabolites (such as succinate and/or acetyl-CoA) and hence reduced HIF1 α levels. This illustrates that mitochondrial pyruvate import via MPC is essential to induce inflammatory responses of lung macrophages, and not BMDM, upon viral infections such as SARS-CoV-2 (Zhu et al, 2023; Ran et al, 2023).”

Additional text in the MPC conclusion paragraph:

“Overall, the role of MPC in sepsis could be twofold. On the one hand, it seems that MPC-mediated pyruvate translocation in the mitochondria is important for inducing macrophage inflammatory responses. However, it should be noted that this effect is dependent on the type of infection (bacterial vs viral) and the specific cell-type (BMDM vs lung macrophages). On the

other hand, MPC activity could potentially contribute to metabolic dysregulations during sepsis, but this warrants further investigation.”

2. (Page 18~21) Insufficient Linkage Between PC-Mediated Metabolic Changes and Inflammation:

The text mentions the importance of PC in intermediary metabolism and its regulation by acetyl-CoA but does not sufficiently link these metabolic changes to inflammation. Include detailed explanations of how the metabolic changes mediated by PC, such as alterations in oxaloacetate levels, impact inflammatory pathways, ROS production, or other cellular responses relevant to sepsis.

Answer: We appreciate the reviewer’s suggestion and we have incorporated studies that demonstrate how PC-mediated metabolic changes have an impact on inflammatory-induced NO synthesis, glutathione antioxidant mechanism, inflammation-induced ROS accumulation and the viral innate immune response, see below.

“Interestingly, pancreatic islet β -cells also exhibit reduced PC activity when treated with pro-inflammatory cytokines (TNF α , IL-1 β and IFN- γ). Increasing PC activity, and thus PC-mediated TCA anaplerosis, protects β -cells against inflammation toxicity by increasing aspartate availability through oxaloacetate conversion via aspartate aminotransferase. Elevated aspartate creates an argininosuccinate shunt, activating the urea cycle and redirecting arginine metabolism towards ureagenesis. This limits arginine use for inflammation-induced NO synthesis, reducing oxidative damage and preventing cell death of pancreatic islet β -cells (Fu et al, 2020). PC activity can also facilitate an extra protective mechanism against oxidative stress by controlling the NADPH/NADP⁺ ratio through sustaining malic enzyme activity. Liver-specific PC knockout (LPCKO) mice show diminished malate production and accumulation of pyruvate, resulting in altered malic enzyme activity leading to NADPH/NADP⁺ depletion, diminished antioxidant defenses via a reduced glutathione antioxidant mechanism, and increased liver inflammation susceptibility (Cappel et al, 2019). In addition to preserving the glutathione antioxidant activity, PC is crucial for de novo glutathione synthesis, thereby limiting inflammation-induced ROS accumulation in pancreatic islet β -cells during inflammatory stress (Fu et al, 2021). Conversely, a recent study by Liang et al. demonstrated increased PC activity in in vitro (LPS-treated colorectal cells and LPS-treated monocytes) and in vivo (Dextran sulfate sodium (DSS)-induced) models of colitis. This was accompanied with elevated NF- κ B signaling, increased production of inflammatory cytokines and ROS, decreased pyruvate content, and accumulation of acetyl-CoA, oxaloacetate and lactate. Moreover, treatment with anemoside B4, a specific PC inhibitor, could ameliorate the inflammatory response, oxidative stress, and could regulate the levels of pyruvate and its downstream metabolites. These findings suggest that PC-mediated changes in pyruvate and acetyl-CoA levels are essential in mediating the inflammatory response and ROS production in LPS-treated colorectal cells. Hence, targeting PC with anemoside B4 might be a potential anti-inflammatory approach (Liang et al, 2024). Notably, PC activity is also crucial in viral innate immune responses by targeting RIG-I-MAVS-TRAF6-signaling pathway. This promotes NF- κ B activation and increases production of interferons and pro-inflammatory cytokines in viral infected cells (Cao et al, 2016).”

3. Lack of Specificity in Describing Metabolic Pathways and Ambiguous Statements on Metabolic Shifts:

The manuscript describes pyruvate metabolism in broad terms, discussing concepts like metabolic shifts and changes in enzyme expression or phosphorylation patterns. However, it does not provide specific details on how these enzyme changes lead to alterations in metabolite levels and how these

metabolic changes, in turn, result in inflammation, cell death, or other outcomes relevant to sepsis. Additionally, the manuscript mentions shifts in metabolic pathways in septic patients but does not provide sufficient data or references to support these claims. Including more detailed descriptions, specific examples, and supporting data or references would enhance the understanding of how altered enzyme activities impact sepsis outcomes. This would provide a clearer link between metabolic changes and the clinical manifestations of sepsis.

Answer: We have made significant revisions to the manuscript to address the comment of the reviewer as follows:

- We have included a comprehensive section, Box 1: *Metabolic alterations and failing starvation response in sepsis*, which provides a detailed explanation (with additional data and references) of the metabolic shifts observed in septic patients for more clarity:

“Box 1: Metabolic alterations & failing starvation response in sepsis

Hallmarks of sepsis include immune activation, phagocytosis, acute phase reactant production, fever, tachycardia, and tachypnea, which all require increased supraphysiological energy supplies (Van Wyngene et al, 2018; Wasyluk & Zwolak, 2021). Despite this, septic patients are often unwilling or unable to consume food and exhibit mitochondrial dysfunction, resulting in a reduced capacity of cells to produce ATP (Peterson et al, 2010; Wang et al, 2016; Singer & Brealey, 1999). Muscle biopsies from septic patients are characterized by a diminished ATP/ADP ratio, correlated with sepsis-induced multiple organ failure and poor outcome (Brealey et al, 2002; Fredriksson et al, 2006). This energy imbalance in septic individuals leads to the activation of a starvation response, as evidenced by the following observations. (1) Adipose tissue in septic patients shows increased lipolysis, resulting in higher blood levels of free fatty acids (FFAs), triglycerides, and glycerol, which are significantly higher in septic non-survivors compared to sepsis survivors (Ilias et al, 2014; Rittig et al, 2016; Langley et al, 2013; Lee et al, 2015; Wang et al, 2020). (2) Hepatic cellular glycogen reserves are heavily depleted (Vandewalle et al, 2021). (3) Skeletal muscle proteolysis is activated, leading to increased concentrations of amino acids (e.g. alanine and glutamine) in the blood (Long et al, 1981; Su et al, 2015; Langley et al, 2013; Wang et al, 2020). Interestingly, many studies show that the conversion of these energy-rich substrates into useful metabolites (e.g. acetyl-CoA, ketone bodies and glucose) is disturbed in septic individuals. A proteomic and metabolomic study on plasma samples of sepsis patients shows decreased protein levels of nine fatty acid transporters but elevated protein levels of two fatty acid-binding proteins in sepsis non-survivors, suggesting a profound defect in fatty acid β -oxidation (Langley et al, 2013, 2014). This defect is caused by impaired functioning of the peroxisome proliferator-activated α receptor (PPAR α) as septic individuals exhibit reduced expression of PPAR α and PPAR α -dependent genes correlating with severity (Wong et al, 2009; Standage et al, 2012; Van Wyngene et al, 2020). This results in harmful FFA accumulation leading to lipotoxicity and contributing to multiple organ damage and failure (Van Wyngene et al, 2020). On the other hand, septic non-survivors show elevated plasma levels of citrate, malate, pyruvate, dihydroxacetone, lactate and gluconeogenic amino acids, suggesting alterations in glycolysis, the TCA cycle and GNEO pathways (Langley et al, 2013, 2014; Wang et al, 2020). Indeed, studies show that GNEO is critical for surviving sepsis but often fails due to several factors. Firstly, the glucocorticoid receptor, which regulates the transcription of genes encoding GNEO enzymes like Pck1 (encoding PEPCK) and G6Pc (encoding G6Pase), becomes dysfunctional in sepsis, leading to impaired GNEO (Vandewalle et al, 2021). Secondly, the production of pro-inflammatory cytokines following exposure to

endotoxic shock decreases the transcription of the rate-limiting enzyme Pck1 by reducing the expression of the nuclear receptor cofactor PGC1a, the latter being an essential cofactor for Pck1 transcription (Chichelnitskiy et al, 2009). Lastly, oxidative inhibition of liver G6Pase driven by iron further contributes to GNEO failure during sepsis. Interestingly, counteracting G6Pase repression using ferritin has been shown to preserve GNEO and reduce sepsis mortality, highlighting the importance of sustaining this process during sepsis (Weis et al, 2017). Altogether, this suggests the presence of a sepsis-associated failing starvation response correlating with a bad outcome.”

- Additionally, we have adapted the text wherever possible to highlight more how specific changes in enzyme activity result in inflammation, cell death, or other outcomes relevant to sepsis, see below:

Pyruvate kinase:

“In sepsis, metabolic reprogramming from OXPHOS to aerobic glycolysis is a typical hallmark of activated immune cells in order to accommodate their high energy requirements for performing their inflammatory responses. Indeed, serum PKM2 levels are increased in sepsis patients and are positively correlated with blood glucose, lactate, LDH and disease severity (Wang et al, 2023). Additionally, urinary PKM2 expression is elevated in patients with sepsis-associated acute kidney injury and positively correlates with serum creatinine levels (Jiajun et al, 2024). Hence, this suggests that septic patients are characterized by a PKM2-mediated glucose metabolic reprogramming, and that PKM2 could potentially be used as a prognostic biomarker. Generally, PKM2 can be induced by hypoxia or after LPS stimulation (Wasyluk & Zwolak, 2021). More specifically, LPS-stimulated macrophages show increased PKM2 expression and simultaneous phosphorylation of PKM2 on Tyrosine 105 keeping PKM2 in its dimeric, inactive form. With this, PKM2 can translocate to the nucleus resulting in the activation of hypoxia-inducible factor 1 α (HIF1 α) via the formation of PKM2-HIF1 α complex (Palsson-Mcdermott et al, 2015). On the one hand, PKM2-mediated activation of HIF1 α leads to a metabolic shift towards aerobic glycolysis by inducing the expression of glycolysis-related genes, resulting in elevated lactate production. In turn, lactate inhibits histone deacetylase activity, resulting in high-mobility box 1 (HMGB1) hyperacetylation and its subsequent release (Yang et al, 2014). Additionally, this PKM2-mediated lactate production induces EIF2AK2 phosphorylation resulting in NLRP3 and AIM2 inflammasome activation and subsequent pro-inflammatory mediator release (IL-1 β , IL-18 and HMGB1) in LPS-treated macrophages (Xie et al, 2016). On the other hand, PKM2-HIF1 α complex can bind to the IL-1 β promoter and induces excessive IL-1 β production in LPS-activated macrophages (Palsson-Mcdermott et al, 2015). Furthermore, these studies indicate that the potential PKM2 inhibitor, shikonin, reduces PKM2 activity in LPS stimulated macrophages. This reduction protects mice from lethal endotoxemia and sepsis by partially decreasing lactate production, and consequently pro-inflammatory cytokines release (Yang et al, 2014; Xie et al, 2016). This study confirms the need for therapeutic interventions addressing PKM2-mediated aerobic glycolysis as a metabolic control in inflammation for the treatment of sepsis.”

“PKM2-mediated protection against SIC is also necessary to attenuate LPS-induced mitochondrial damage marked by diminished ATP production, reduced mitochondrial respiratory complex I/III activities and increased ROS production, eventually resulting in enhanced myocardial inflammation and impaired cardiac function. This is mediated via PKM2-dependent phosphorylation of prohibitin 2 (essential for the preservation of mitochondrial function and structure) in order to limit LPS-mediated prohibitin 2 degradation (Ren et al, 2024; Du et al, 2024). Hence, cardiomyocyte-specific PKM2 overexpression exhibits

protective effects against SIC in mice with LPS or gram-negative bacteria induced sepsis (Ni et al, 2022; Du et al, 2024)."

Lactate dehydrogenase:

To provide more clarity regarding the effects of lactate, an additional text in the manuscript was provided along with an additional section: Box 2: *The versatile role of lactate in sepsis*, see below.

"This protective effect results from the role of lactate in mediating HMGB1 lactylation via its direct uptake through monocarboxylate transporters (MCTs) or via a p300/CBP dependent mechanism in activated macrophages. Lactate can also bind to its receptor, G-protein coupled receptor 81 (GPR81), leading to decreased expression of SIRT1 deacetylase, thereby inducing increased HMGB1 acetylation in activated macrophages. Eventually, lactylation and acetylation of HMGB1 lead to its exosomal secretion in macrophages, contributing to endothelial barrier dysfunction and vascular permeability which exacerbates sepsis progression"

"The versatile role of lactate in sepsis"

For decades, lactate was considered as an inert by-product of glycolysis rather than a bioactive molecule. Interestingly, lactate can also function as a signaling molecule and can be transported into cells by the monocarboxylate transporters (MCTs), which play a crucial role in metabolic communication between cells. MCT4 is specialized for lactate export, while MCT1 can either import or export lactate, depending on whether it is expressed by oxidative or glycolytic cells.

Lactate is often considered harmful in sepsis; Injection in septic animals worsens sepsis, whereas inhibiting lactate production with the LDH inhibitor, oxamate, improves sepsis survival (Yang et al, 2022). In macrophages, extracellular lactate uptake leads to HMGB1 lactylation, promoting its release and increasing endothelial permeability (Yang et al, 2022). Inhibiting lactate uptake with CHC, an MCT inhibitor, suppresses lactate-induced HMGB1 lactylation in macrophages. In neutrophils, lactate uptake through MCT1 upregulates PD-L1 expression, delaying apoptosis in these cells. Administration of the selective MCT1 inhibitor AZD3965 increases neutrophil apoptosis leading to enhanced survival in CLP mice (Fei et al, 2024). Research by Vandewalle et al. even demonstrates that lactate significantly contributes to sepsis lethality as hyperlactatemia, in combination with a sepsis-induced glucocorticoid resistance, results in a lethal vascular collapse via uncontrolled vascular endothelial barrier dysfunction (Vandewalle et al, 2021).

Contrary to these findings, other researchers have reported that lactate infusion can improve sepsis outcomes by enhancing hemodynamics and reducing inflammation (Walenta et al, 2000). Lactate's role in immunosuppression is well-documented in cancer research and also in inflammatory conditions, where it has been shown to reduce organ damage (Walenta et al, 2000; Hoque et al, 2014). Recently, lactate produced by monocyte-derived human tolerogenic dendritic cells has been found to decrease T cell proliferation, delaying graft-versus-host disease (Marin et al, 2019).

Lactate also has direct protective effects on organs such as the heart and brain. In the heart, it serves as an energy source, and pharmacological inhibition of MCT4—which blocks lactate export—mitigates heart failure in mice. Systemic lactate deprivation using dichloroacetate and ICI-118551 (inhibiting respectively PDK and β 2-adrenergic receptors) is linked to

decreased myocardial energetics, reduced cardiovascular performance, and early death in endotoxic shock (Levy et al, 2007). In the brain, lactate compensates for reduced glucose uptake following traumatic brain injury (Glenn et al, 2015). Given the starvation response observed in sepsis patients, it stands to reason that lactate is necessary for maintaining myocardial and cerebral metabolism during sepsis (Vandewalle & Libert, 2022).

Additionally, lactate can directly affect pathogens. For example, Candida albicans utilizes lactate to increase its resistance to host-relevant stressors, enhance biofilm formation, and evade macrophage recognition (Williams & Lorenz). Conversely, lactate inhibits the growth of Salmonella in vitro and reduces mortality from Salmonella infection in vivo (Iraporda et al, 2017).

In summary, lactate acts as a double-edged sword, with its effects varying based on dose, type (lactic acid versus lactate anion), metabolic state, cell type, and pathology. While high lactate levels are clearly detrimental in sepsis, some clinicians now recognize that mild lactate production during sepsis is a compensatory mechanism. The so-called "Lactate Shuttle Theory" leverages lactate as a signaling and energy molecule to meet increased energy demands and dampen the inflammatory response during sepsis (Brooks, 2018).

Of note, sepsis is frequently associated with lactic acidosis, which is typically defined as a blood lactate concentration exceeding 5 mM combined with a blood pH below 7.35 (Mizock, 1992). The precise role of glycolysis in inducing lactatemia or lactic acidosis remains unclear. During glycolysis, one molecule of glucose yields two molecules of pyruvate, 2 ATP, 2 NADH, and 2 protons (H^+). When pyruvate is converted to lactate by LDH, NADH is reverted to NAD^+ , consuming an equivalent amount of protons. Thus, an increase in lactate production leading to hyperlactatemia is not inherently acidifying. The primary source of protons, and hence acid, is ATP hydrolysis. The Krebs cycle actively consumes these protons, so a reduction in Krebs cycle flux due to impaired oxygen utilization or mitochondrial dysfunction leads to acid accumulation. Simultaneously, reduced Krebs cycle activity results in lactate production. The cotransport of lactate and protons by MCT causes tissue hypoxia-related acidosis to manifest as "lactic acidosis" clinically. Therefore, the term "lactic acidosis" is a misnomer, and "lactate-associated acidosis" is more accurate, as lactate itself does not cause acidosis (Müller et al, 2023). In sepsis, this lactate-associated acidosis is primarily compensated for by the kidneys, which decrease strong anions, thus widening the strong ion difference. Consequently, the severity of acidemia heavily depends on renal function (Gattinoni et al, 2019)."

Pyruvate carboxylase:

We would like to refer to our answer to comment 2 - referee #3.

Alanine aminotransferase, mitochondrial pyruvate carrier and pyruvate dehydrogenase:

As research towards the role of ALT and MPC in sepsis individuals is currently limited, we could not provide additional insights or references regarding the role of ALT or MPC-induced changes in metabolite levels and inflammation. Additionally, the effects of PDC activity are already extensively described in the manuscript and due to word limit constraints and the risk of overloading the manuscript, no additional insights could be provided. We believe that the current level of detail is sufficient to convey the critical role of these enzymes in the context of sepsis while maintaining the overall clarity and focus of the manuscript.

4. Insufficient Integration of Clinical Data:

The manuscript discusses several metabolic pathways and therapeutic strategies without adequately considering their clinical applicability. Additionally, it does not integrate clinical data effectively to support its claims about metabolic interventions in sepsis. Discussing the current state of therapeutic development based on these metabolic insights would enhance understanding of how these changes translate into treatments and their impact on patient outcomes. Given the preclinical nature of the results, it may be advisable to adjust the title to better reflect the scope, such as emphasizing "mechanisms and potential therapeutic targets" rather than "therapeutic approaches."

Answer: This is a good comment of the reviewer. We have revised the text to emphasize human data wherever possible. However, it should be noted that clinical research in sepsis patients poses significant challenges and is restricted. In addition, current research on the roles of ALT, MPC and PC in sepsis models is notably limited, and to our knowledge, there are no clinical observations available for these enzymes in sepsis patients. We have therefore highlighted the need for more clinical sepsis research in the "Pending Issues" section. Despite these limitations, we have focused to cite human data regarding changes in PK, LDH and PDH as these enzymes are more commonly studied in sepsis research. Furthermore, due to word limit constraints and to stay within the scope of this review, it was not possible to include a discussion regarding the major differences between human sepsis and mouse model of sepsis.

Please see the revised sections below:

- Pyruvate kinase: We have cited human data (see below) showing increased serum PKM2 levels that correlate with glucose, lactate, LDH and disease severity. This is followed by mouse data that provide mechanistic insights into PKM2-mediated aerobic glycolysis, its consequences and potential therapeutic strategies

"In sepsis, metabolic reprogramming from OXPHOS to aerobic glycolysis is a typical hallmark of activated immune cells in order to accommodate their high energy requirements for performing their inflammatory responses. Indeed, serum PKM2 levels are increased in sepsis patients and are positively correlated with blood glucose, lactate, LDH and disease severity (Wang et al, 2023). Additionally, urinary PKM2 expression is elevated in patients with sepsis-associated acute kidney injury, positively correlated with serum creatinine levels (Jiajun et al, 2024). Hence, this suggests that septic patients are characterized by a PKM2-mediated glucose metabolic reprogramming, and that PKM2 could potentially be used as a prognostic biomarker."

- Lactate dehydrogenase: We have added additional human data, along with in vitro studies, showing increased serum LDH levels correlated with higher lactate, IL-1b and mortality as evidence for increased LDH activity in septic subjects. Additionally, increased levels of lactate in the skeletal muscle of septic shock patients are observed by Levy et al. indicating increased LDHA activity in the skeletal muscle of sepsis patients. After this, the protection mechanism by therapeutically targeting LDHA activity in sepsis models is discussed.

"Indeed, several clinical studies have provided evidence that serum lactate levels are elevated and may hold significance in predicting mortality in patients with septic shock (defined by Sepsis-3) (Ryoo et al, 2018; Lee et al, 2021). Moreover, sepsis patients show increased serum LDH levels which is positively correlated with serum lactate, IL-1b and 28-day mortality, thereby suggesting that glucose metabolic

reprogramming of immune cells contributes to sepsis mortality (Lu et al, 2018; Frenkel et al, 2023). Moreover, many in vitro studies show increased levels of LDHA upon LPS stimulation in macrophages (Yang et al, 2014; Palsson-Mcdermott et al, 2015; Xie et al, 2016; Zhang et al, 2022). Apart from lactate production by immune cells, Levy et al. observe elevated levels of pyruvate and lactate in the skeletal muscle of septic shock patients. Interestingly, this is driven by the release of epinephrine which triggers Na^+K^+ ATPase activity to induce membrane hyperpolarization. This increased activity of the Na^+K^+ pump necessitates greater ATP consumption, which is facilitated by an enhanced aerobic glycolysis potential, and thus increased LDH activity. Consequently, this process is associated with Na^+K^+ ATPase-linked lactate production in skeletal muscle, as evidenced by the administration of a specific Na^+K^+ ATPase inhibitor, ouabain, which significantly reduces skeletal muscle pyruvate and lactate levels in patients with septic shock (Levy et al, 2005). Altogether, this highlights the need for therapeutic interventions targeting LDHA in sepsis.”

- Pyruvate dehydrogenase: We have mentioned one clinical study regarding impaired PDC activity, as well as two DCA clinical trials that demonstrated reduced lactate levels with DCA administration but no survival benefit. However, despite the findings of Stacpoole et al., many studies in murine sepsis models have shown improved survival with DCA treatment (Stacpoole et al, 1992). Given that only one clinical study reported no survival benefit and no other clinical studies with DCA have been performed, further clinical trials investigating DCA are still worth pursuing and may require optimization.

“Moreover, peripheral blood mononuclear cells of septic patients exhibit diminished PDC levels and activity compared to healthy controls which was associated with reduced survival chance (Nuzzo et al, 2015). Interestingly, targeting PDK with the prototypic drug, dichloroacetate (DCA, a structural analogue of pyruvate), lowers phosphorylation on the E1 α subunit which ultimately improves PDH activity and lowers blood lactate levels. Indeed, DCA treatment improves oxygen consumption, pyruvate oxidation and reduces plasma lactate concentration in septic patients (Stacpoole et al, 1992; Gore & Demaria, 1996). However, the clinical trial of Stacpoole et al. shows that DCA treatment fails to improve the survival rate (Stacpoole et al, 1992). Despite this, DCA protection has been widely observed in many studies using sepsis models.”

Additional text in the PDC conclusion paragraph:

“However, clinical trials studying the efficacy of DCA in septic patients remain limited. DCA treatment could effectively reduce hyperlactatemia, but fails to improve the survival rates (Stacpoole et al, 1992). Nevertheless, further exploration of the protective role of DCA in sepsis patients is strongly recommended.”

The last paragraphs of every subtitle provide an overview of the current state of therapeutic development and the potential future perspectives, when possible.

Additionally, we have adapted the title of the manuscript to “Sepsis-induced changes in pyruvate metabolism: insights and potential therapeutic approaches.” The abstract is slightly adapted to “Based on the available data, we also discuss potential therapeutic strategies targeting these pyruvate-related enzymes leading to enhanced survival.”

Minor points:

5. (Page 2) Inaccurate Statement in the abstract

The abstract states, "In mammals, pyruvate is the only substrate for lactate production," which seems too strong. It might be more precise to state that while pyruvate is the primary substrate for lactate production, other metabolic intermediates like alanine can be converted to pyruvate and subsequently to lactate. This would better reflect the metabolic flexibility and integration of pathways in cellular energy metabolism.

Answer: We have adapted the text in the abstract according to the reviewers suggestion, see below:

"In mammals, pyruvate is the primary substrate for lactate production."

6. (Page 3) Misleading Statement on Starvation Response:

The manuscript states that septic patients trigger a starvation response due to an energy deficit but does not provide sufficient evidence or explanation for this claim. It is recommended to provide more context and references to support the claim that a starvation response is activated in septic patients due to energy deficits, or to discuss other factors that may contribute to the energy deficit. Additionally, more explanation is needed on the sepsis-associated failing starvation process to clarify its role and implications in the context of sepsis. This added detail would enhance the understanding of metabolic responses in septic patients.

Answer: We would like to refer to our answer (Box1: *Metabolic alterations and failing starvation response in sepsis*) to comment 3 from referee #3. Additionally, we have added the starvation response definition to the glossary for more clarity.

7. (Page 8) Insufficient Detail on Enzyme Activities (Page 8)

The discussion on the role of various enzymes, such as PKM2 and LDHA, in sepsis is not detailed enough. Include more detailed explanations of how these enzymes are regulated in sepsis and their specific contributions to metabolic dysfunctions.

Answer: This is a good suggestion of the reviewer and we have adapted the text to give a more detailed explanation of how PKM2 and LDHA are regulated in sepsis and how they contribute to metabolic dysfunctions, see below.

Pyruvate kinase:

We would like to refer to our answer (Pyruvate kinase) to comment 3 from referee #3.

Lactate dehydrogenase:

We would like to refer to our answer (Lactate dehydrogenase) to comment 3 from referee #3 and the additional text below:

"As mentioned in the previous section, sepsis is associated with glucose metabolic reprogramming of immune cells (i.e. Warburg phenotype) potentially mediated via elevated HIF1 α activation. Apart from PKM2 and hypoxic conditions, bacterial products (e.g. LPS via mTOR activation) and pro-inflammatory cytokines (e.g. TNF α) can induce HIF1 α activity under normoxic conditions (Nishi et al, 2008; Rigueira et al, 2009). In turn, HIF1 α induces the expression of glycolysis-related genes including LDH, thereby increasing the aerobic glycolytic flux responsible for elevated pyruvate levels driving LDH activity in producing lactate (Garcia-Alvarez et al, 2014; TAN et al, 2021). Indeed, several clinical studies have provided evidence

that serum lactate levels are elevated and may hold significance in predicting mortality in patients with septic shock (defined by Sepsis-3) (Ryoo et al, 2018; Lee et al, 2021). Moreover, sepsis patients show increased serum LDH levels which is positively correlated with serum lactate, IL-1 β and 28-day mortality, thereby suggesting that glucose metabolic reprogramming of immune cells contributes to sepsis mortality (Lu et al, 2018; Frenkel et al, 2023). Moreover, many in vitro studies show increased levels of LDHA upon LPS stimulation in macrophages (Yang et al, 2014; Palsson-Mcdermott et al, 2015; Xie et al, 2016; Zhang et al, 2022)”

8. (Page 15) Clarification on PDH Complex Terminology:

The manuscript describes "PDH" as a large multi-component enzymatic complex comprising various catalytic, binding, and regulatory components. However, it would be more accurate to refer to this as the "PDH complex" (PDC) rather than simply "PDH." For example, on page 15, the first line should state "PDH complex is a large multi-component enzymatic complex (10 MDa) comprising three catalytic, binding, and regulatory components" instead of "PDH is a large multi-component enzymatic complex." It is important to consistently and clearly use the terms "PDH" and "PDH complex (PDC)" throughout the manuscript to avoid confusion. This clarification will help ensure precise communication of the enzymatic components and their functions.

Answer: Indeed, we agree with the reviewer to clarify the terminology used when referring to pyruvate dehydrogenase (PDH) complex. We have adapted the abbreviation of PDH complex to PDC and changed the PDH abbreviation to PDC throughout the manuscript when applicable. We maintained the PDH abbreviation when the text specifically mentions phosphorylation of the PDH component of PDC. The abbreviation of PDH complex was correctly changed to PDC on Figures 2, 4 and 6.

9. (Page 16) Comprehensive Discussion on PDK Isoform Activities:

Based on the review of the Sugden & Holness (2006) reference, it is evident that PDK4 exhibits higher activity towards phosphorylation at site 2 compared with PDK1, PDK2, and PDK3. To provide a more comprehensive understanding of the regulation of PDH, I recommend including a discussion on the distinct phosphorylation site preferences and activities of all four PDK isoforms, with a mention of the significant role of PDK4. This addition would enrich the analysis by illustrating how the differential regulation by PDK isoforms influences PDH activity and, consequently, affects cellular energy metabolism in various physiological and pathological contexts.

Answer: We understand this comment of the reviewer and we have revised the manuscript to include a more detailed explanation of the phosphorylation site preferences of the different PDKs, see below:

“Furthermore, all four isoforms exhibit distinct preferences and phosphorylation rates at the three sites of PDH. PDK1 can phosphorylate all three sites with a preference order of site 1 > site 3 > site 2. In contrast, the other PDKs (PDK2-4) only phosphorylate site 1 and 2 of PDH. PDK2 shows a preference for phosphorylating site 1 over site 2. Therefore, PDK2 has less phosphorylation capacity for site 2 compared to PDK3 and PDK4, with PDK4 displaying the highest activity towards site 2. PDK3 phosphorylates site 1 faster than site 2. Overall, the phosphorylation potential of PDH by all PDKs ranks as follows: PDK1 > PDK3&4 > PDK2 (Korotchkina & Patel, 2001; Kolobova et al, 2001).”

10. (Page 22) Redundant Information

There is redundant information about the role of PDK and thiamine in pyruvate metabolism and sepsis. Consolidate this information to avoid repetition and improve the flow of the manuscript.

Answer: We have adapted the text wherever possible to ensure a better flow of the manuscript.

11. (Page 26) Oversimplified Statement on Pyruvate Metabolism:

The statement "The formed cytosolic pyruvate will be directly metabolized into lactate" could be seen as oversimplifying the process. It would be more accurate to state that "the formed cytosolic pyruvate is predominantly converted into lactate by lactate dehydrogenase," acknowledging that not all pyruvate is converted to lactate.

Answer: We understand this suggestion of the reviewer and we have adapted the legend of Figure 1, see below:

"The formed cytosolic pyruvate is predominantly converted into lactate by lactate dehydrogenase (LDH) thereby regenerating NAD^+ from NADH for maintaining glycolysis activity"

12. Clarification on Abbreviation :

The manuscript uses the abbreviation "GNEO," which seems to stand for "gluconeogenesis," but there is no information provided about this abbreviation. It is recommended to clearly define "GNEO" as "gluconeogenesis" upon its first use to avoid any confusion and ensure clarity for the readers. This clarification will help in understanding the context and maintaining consistency throughout the manuscript.

Answer: Thank you for pointing this out. Indeed, "GNEO" stands for gluconeogenesis. We have now revised the text to clearly indicate that GNEO refers to gluconeogenesis, and we have also added this abbreviation to the gluconeogenesis definition in the glossary.

13. (page 26) Omission of Key Pathways:

The description omits the role of pyruvate in fatty acid synthesis. Pyruvate can be converted to acetyl-CoA, which can be used for fatty acid synthesis in the cytosol after being exported out of the mitochondria as citrate and then converted back to acetyl-CoA in the cytosol. Additionally, the conversion of pyruvate to malate by malic enzyme produces NADPH, which is crucial for steroid synthesis and other anabolic processes. Including these pathways would provide a more comprehensive overview of pyruvate metabolism, highlighting its role in lipid biosynthesis and related anabolic routes.

Answer: This is a good comment of the reviewer and we have adapted the manuscript to provide a more comprehensive overview of pyruvate metabolism, see below. Additionally, de novo fatty acid synthesis and malic enzyme are added on Figure 2.

"Depending on the energy status of the cell, the transported pyruvate can either be oxidized by pyruvate dehydrogenase (PDH) to form carbon dioxide (CO_2) and acetyl-CoA, which can be further oxidized by the TCA cycle for mitochondrial ATP generation, or used for fatty acid synthesis. Alternatively, the transported pyruvate can be carboxylated by pyruvate carboxylase (PC) to generate oxaloacetate which can either drive the TCA cycle, fatty acid synthesis or it can be used in pyruvate-driven gluconeogenesis (GNEO). Another important route for pyruvate metabolism involves malic enzyme (ME), which catalyzes the oxidative

decarboxylation of malate to pyruvate and CO₂ (Gray et al, 2014; Jeoung et al, 2014; Prochownik & Wang, 2021) (Figure 2)."

An additional sentence was added to the introduction paragraph of pyruvate dehydrogenase:

"Additionally, PDC activity is also crucial for providing acetyl-CoA for FFA synthesis (Sugden & Holness, 2006)."

An additional paragraph (2.3) of malic enzyme was included in the manuscript to provide some explanation of this enzyme

"2.3 Malic enzyme"

MEs are oxidoreductases that catalyze the oxidative decarboxylation of L-malate to pyruvate and CO₂ with the simultaneous reduction of NAD(P)⁺ to NAD(P)H. It functions as a metabolic node connecting glycolysis with the TCA cycle, and it is also important in glutamine metabolism and GNEO. Moreover, NADPH production is essential for lipid biosynthesis and redox homeostasis (Prochownik & Wang, 2021). In mammals, three different ME isoforms have been identified: (1) cytosolic, NADP⁺-dependent ME (ME1), mitochondrial NADP⁺-dependent ME (ME2), and mitochondrial NADP⁺-dependent ME (ME3). This distinction is made based on subcellular localization and cofactor use. Each isoform is a homotetrameric protein with a double dimer structure, wherein the dimer interface promotes a more robust interaction compared to the tetramer interface. Each monomer is composed out of four structural domains (A,B,C and D) and the active site of the enzyme lays on the interface between domains B and C (Chang & Tong, 2003). Furthermore, the activity of MEs is mainly controlled by transcriptional regulation, post-translational modifications such as phosphorylation and acetylation, and allosteric regulation with fumarate as an activator and ATP as an inhibitor (Chang & Tong, 2003). However, studies towards the role of ME activity in sepsis are currently limited. Despite this, it should be insightful to investigate potential disturbances in ME activity in septic individuals and its potential role in sepsis metabolic alterations, such as impaired GNEO resulting in hypoglycemia."

14. (Page 27) Clarification on Glutamate and the Urea Cycle:

The statement "The formed glutamate enters the urea cycle to eliminate its amino group" should be clarified. The amino group from glutamate is removed as ammonia, which then enters the urea cycle. The direct entry of glutamate into the urea cycle is not accurate; it is the ammonia derived from glutamate that is converted to urea. Including this clarification would provide a more precise explanation of the metabolic pathway.

Answer: We agree with the reviewer and we included this clarification in the manuscript, see below.

Manuscript:

"The alanine cycle shows close similarity with the Cori cycle, but it is less efficient as deamination of glutamate releases its amino group as ammonia, which needs to be detoxified via the urea cycle"

Legend Figure 3:

"The formed glutamate is deaminated resulting in the release of its amino group as ammonia, which is then eliminated via the urea cycle"

References

- Batenburg JJ & Olson MS (1976) Regulation of pyruvate dehydrogenase by fatty acid in isolated rat liver mitochondria. *J Biol Chem* 251: 1364–1370
- Bradley NS, Heigenhauser GJF, Roy BD, Staples EM, Inglis JG, LeBlanc PJ & Peters SJ (2008) The acute effects of differential dietary fatty acids on human skeletal muscle pyruvate dehydrogenase activity. *J Appl Physiol* 104: 1–9
- Stacpoole PW, Wright EC, Baumgartner TG, Bersin RM, Bughalter S & Curry SH (1992) A controlled clinical trial of dichloroacetate for treatment of lactic acidosis in adults. 326
- Van Wyngene L, Vanderhaeghen T, Timmermans S, Vandewalle J, Van Looveren K, Souffriau J, Wallaey C, Eggermont M, Ernst S, Van Hamme E, *et al* (2020) Hepatic PPAR α function and lipid metabolic pathways are dysregulated in polymicrobial sepsis. *EMBO Mol Med* 12: 1–20
- Zhou Y & Grill VE (1995) Palmitate-Induced B-cell insensitivity to glucose is coupled to decreased pyruvate dehydrogenase activity and enhanced kinase activity in rat pancreatic islets. 44
- Zhou YP, Berggren PO & Grill V (1996) A fatty acid-induced decrease in pyruvate dehydrogenase activity is an important determinant of β -cell dysfunction in the obese diabetic db/db mouse. *Diabetes* 45: 580–586

18th Sep 2024

Dear Claude,

Thank you for the submission of your revised manuscript to EMBO Molecular Medicine. We have now received feedback from the experts who agreed to evaluate your manuscript, and as you will see from the reports below, they are satisfied with the revisions and support publication. I will therefore be able to accept your manuscript once the following editorial issues are addressed:

- Please provide up to 5 keywords
- Acknowledgements: funding information provided here should match information entered in the submission system, please adjust accordingly.
- Please add the 'Disclosure and competing interests' statement to your manuscript file.
- Thank you for providing an extensive glossary. Please add it to the manuscript file.
- Please also add the Pending issues to the manuscript file.
- Your figures have been forwarded to our scientific illustrator who will contact you for approval. Please carefully check the redrawn figures once you receive them for accuracy.

Looking forward to receiving your revised manuscript,

With kind regards,

Lise

***** Reviewer's comments *****

Referee #1 (Remarks for Author):

none

Referee #2 (Remarks for Author):

The authors have now satisfactorily addressed my suggestions and concerns.

Referee #3 (Remarks for Author):

The revised manuscript shows significant improvement in both depth of analysis and clarity. The additional details on the regulation of metabolic pathways, especially regarding MPC, PC-mediated changes, and enzyme activities, address the previous concerns. The revised structure avoids redundancy and simplifies complex ideas, making it more accessible without losing scientific rigor. Overall, the manuscript is now much more comprehensive and presents a clear, well-supported narrative on pyruvate metabolism in sepsis.

The authors have addressed all minor editorial requests.

26th Sep 2024

Dear Prof. Libert,

Thank you for submitting your revised files. I am pleased to inform you that your manuscript is now accepted for publication and will be sent to our publisher once the figures will be redrawn and approved by you.

Your manuscript will be processed for publication by EMBO Press. It will be copy edited and you will receive page proofs prior to publication. Please note that you will be contacted by Springer Nature Author Services to complete licensing information.

There is no charge for this Review Article, but in a few weeks when you are contacted to sign your license agreement and review article proofs, please enter this token into the appropriate field in the Springer Nature author services system: OTGZnja0NDI5.

If you have any questions, please do not hesitate to contact the Editorial Office. Thank you for your nice contribution to EMBO Molecular Medicine!

With kind regards,

Lise
